



# Analytical approaches for wave energy dissipation induced by wave-generated turbulence and random wave-breaking

Yongzeng Yang [1,3], Fuwei Wang [2], Meng Sun [1,3], Xingjie Jiang [1,3], Xunqiang Yin [1,3], Yongfang Shi [1,3], and Yong Teng [1]

[1] First Institute of Oceanography, Ministry of Natural Resources, Qingdao, Shandong, China

[2] Frontier Science Center for Deep Ocean Multispheres and Earth System (FDOMES) and Physical Oceanography Laboratory, Ocean University of China, Qingdao, Shandong, China

[3] Key Laboratory of Marine Science and Numerical Modeling (MASNUM), Ministry of Natural Resources, Qingdao, Shandong, China.

**Correspondence**: Meng Sun (sunm@fio.org.cn)

**Abstract.** This paper is dedicated to investigate the dissipation effects of wave-generated turbulence reacting on ocean waves and to estimate the energy loss due to wave-breaking theoretically. An analytical dissipation source function induced by wave-generated turbulence in the water was proposed through the equilibrium solutions of a high-deterministic second-order turbulence closure model between the wave shear instability generations and the turbulent kinetic energy (TKE) dissipations. And an improved postbreaking spectrum expression, based on the breaking wave statistical method, was presented to depict more explicitly the intermittent wave-breaking events. Their verifications with laboratory observations or comprehensive measurements were provided, and applications on simple duration-limited growth and decay experiments were implemented. Numerical results indicate that the wave energy dissipation induced only by random wave-breaking is inadequate, sum of its contribution and that induced by wave-generated turbulence play critical roles of wave energy loss. Evaluations for more complex situations will be addressed in the future series of papers.



## 1 Introduction

The third generation ocean wave models, which integrate the dynamical equations that
describe the evolution of a wave field, have been widely used in scientific studies and
practical applications. Knowledge of the other source terms is incomplete but still rational. In
contrast, the least understood aspect of the physics of wave model is the dissipation terms
(Donelan and Yuan, 1994; Young and Babanin, 2006; Babanin, 2011). The dissipation source
(sink) terms induced by wave-generated turbulence and random wave-breaking are the
subject of the present paper. There are a number of dissipation mechanisms studied
previously through either experimental or analytical (or both) approaches, and they have
advanced very significantly over the past decades. However, there remains a notable gap in
mechanism research pertaining to varied parameterizations and applications in wave models.

Concerning to the wave-breaking dissipation, the most mathematically well-advanced and
most frequently utilized whitecap model is that of Hasselmann (1974) which involves the
whitecaps are weak-in-the-mean and the dissipation is a linear function of the spectrum, and
its parameterization has been proposed and further extended in various WAM-Cycle models
by Komen et al. (1984), WAMDI G. (1988), Bidlot et al. (2005), Bidlot (2012), etc.
Following the quasi-saturated ideas of Phillips (1985) and numerical modeling framework of
Alves and Banner (2003), Ardhuin et al. (2009, 2010), Filipot and Ardhuin (2012) presented
an improved wave-breaking dissipation parameterization as the sum of the saturation-based
term and a cumulative breaking term, while the latter represents the smoothing of the surface
by big breakers that wipe out smaller waves. The probability model due to wave breaking
was proposed by Longuet-Higgins (1969), Yuan et al (1986) and Hua and Yuan (1992). This
kind of wave-breaking dissipation source function was derived from the breaking wave
statistics, in which the power of the normalized integral wave steepness parameter is 1/2 and
less than that of whitecap model. So the probability model gives lower dissipation value for
the high sea state condition (Yuan et al., 1993), and has been applied in the LAGFD
(Laboratory of Geophysical Fluid Dynamics)-WAM wave model and MASNUM (Key
Laboratory of MArine Science and NUmerical Modeling) wave model (Yuan et al., 1991;
Yang et al., 2005). Though conceptually attractive, its practical parameterization fitted to the
whitecap model is deprived of its own discretionary estimates of the dissipation rate. Young
and Babanin (2006) summarized series of previous experimental attempts to obtain an
experimental spectral dissipation function, and based on direct estimates of wave energy loss
due to dominant breaking through their comprehensive measurement records at Lake George,



they proposed a new spectral dissipation source term due to wave breaking. This

experimental parameterization was improved and employed in WW3 (WAVEWATCH ⅠⅠⅠ)

wave model (Babanin et al., 2007, 2010; Babanin, 2009; Rogers et al., 2012; Liu et al., 2019;

WW3DG, 2019). The quantitative match is dubious between the latter applications and the

former experimental estimates, despite only a single field record was analyzed but verified

approximately by turbulent kinetic energy (TKE) dissipation rates which were retrieved from

synchronously ADV measured turbulence spectra (Young and Babanin, 2006). In addition,

the physical meaning of wave-breaking dissipation rate (time derivative of wave energy

spectrum), as stated in previous studies, remains fuzzy and vague because the wave-breaking

process is not continuous but very short and intermittent, which stimulate us to propose an

improved analytical postbreaking spectrum expression, from the point of view of the

probability theory, for practical implementation. This constitutes one of our primary focal

points of this study.

Polnikov (1994, 2005, 2010), Polnikov et al. (2006) argued that the mechanism of wave

energy dissipation is completely conditioned by the interaction between wave motions and

the turbulence of the upper water layer, and the latter is generated by a great number of

physical processes, including different kinds of hydrodynamic instabilities of mechanical

motions near the interface. They solved Reynolds' equation where the Reynolds' stress was

expanded into a series with respect to velocity components and their spatial derivatives. The

Prandtl mixing-length hypothesis was used to close the turbulent terms in these series. Their

physical treatment is attractive, but the theory needs further development (Young and

Babanin, 2006), even though it was further constructed by means of the phenomenological

similarity method and spectrally justified in the frame of the proposed eddy viscosity model

(Polnikov, 2012). Tolman and Chalikov (1996) also suggested that the wave energy

dissipation is due to turbulent viscosity. For dominant low-frequency dissipation, if wave

motion and turbulence are not correlated, their interaction can be accounted for by

introducing an effective, although weak, turbulent viscosity coefficient in the oceanic

boundary layer. But for poorly understood high-frequency dissipation, a diagnostic high-

frequency dissipation was defined and designed to result in a consistent source term balance.

And the total dissipation source term was defined as a linear combination of the above high

and low frequency constituents. Their separation parameterization for the dissipation term is

physically meaningful and the heuristic arguments have been widely applied in many

practical numerical models, but the frequency interval is divided arbitrarily into three



portions and the dissipation mechanism is different in different domains. In fact, during the past two decades, a series of studies concerned with mechanisms of nonbreaking wave-generated turbulence and its mixing effects on upper ocean layers have been achieved

significantly and their conclusions have been confirmed more exhaustively by experiments. The theoretical wave-generated turbulence theory can be established by two different but relatively consistent approaches: the first approach involves the use of a parameterization form similar to the classical Prandtl mixing-length theory, and the second approach involves the use of the equilibrium solutions of a high-deterministic second-order turbulence closure

model between the wave motion shear instability generations and the TKE dissipations (Baumert et al., 2005), then the analytical mixing coefficients were proposed to elaborate the dominant mixing intensity induced by wave-generated turbulence in the upper ocean (Yuan et al., 1999; Yuan et al., 2013). Qualitative and quantitative validations by field measurements and improvements of wave-current coupled modelling indicate the key mixing role in the

formation of upper mixed layers (Qiao et al., 2004, 2010; Xia et al., 2004, 2006; Shu et al., 2011; Shi et al., 2016, 2019; Yu et al., 2020; Zhuang et al., 2020, 2021, 2022; Yang et al., 2003, 2004, 2019, 2022). But how the reaction of wave-generated turbulence on ocean waves was disregarded in their studies, which is still lack and needs to investigate further. Babanin (2006), Babanin and Haus (2009), Dai et al. (2010) tested and confirmed the nonbreaking

wave-generated turbulence through mechanically generated laboratory wave experiments. Babanin and Chalikov (2012) also presented that the vorticity and turbulence usually occur in vicinity of wave crests and then spread over upwind slope and downward through a numerical wave-turbulence model. Based on their experimental approximations, a parameterization of swell dissipation rate was proposed, verified through altimeter

observation data and employed in WW3 ST6 package (Babanin 2011, 2012; Young et al., 2013; Zieger et al., 2015; WW3DG, 2019). In fact, an analytical parameterization for wave energy dissipation can be effectively deduced through the equilibrium solutions of wave-generated turbulence, which will be described in detail and implemented in numerical experiments below. It constitutes the other focal point of this study to illustrate the important

role of dissipation induced by wave-generated turbulence, which is definitely the feedback of imparting of wave shear instability generations on turbulence.

There are also a number of other dissipation mechanisms which are certainly not negligible in the wave system. Prominent negative input source term for swell was introduced by Chalikov & Belevich (1993), Chalikov (1995) , Tolman & Chalikov (1996) , Tolman





(2002) and Chalikov & Babanin (2009) where the phase velocity of waves is larger than the
wind velocity, which means that the dynamic pressure of the wind on the forward face of the
wave component exceeds the pressure on the backward face and waves accelerate wind,
resulting in the momentum and energy fluxes from the waves to the wind. Based on the direct
measurements of turbulent air-sea fluxes obtained during several sea expeditions, Grachev

and Fairall (2001) verified that long ocean waves (swell) traveling faster than local wind and
in the same direction cause upward momentum transport, implying a negative drag
coefficient. A weak damping of swells was also introduced by Janssen (2004), who proposed
an asymptotic linearization of the small effects of air turbulent eddies. Ardhuin et al. (2009,
2010) , Collard et al. (2009) proposed a nonlinear swell dissipation parameterization, which is

related to a laminar-to-turbulent transition of the oscillatory boundary layer over swells, using
the spaceborne SAR observed swell fields from the European Space Agency's (ESA)
ENVISAT satellite. Interaction of ocean waves and upper ocean turbulence, while the latter is
induced by Stokes drift shears, accounts for a significant fraction of the energy losses of the
wave field (McWilliams et al., 1997; Teixeira and Belcher, 2002; Ardhuin and Jenkins, 2006).

Evaluations indicated it is much weaker than other dissipations (Ardhuin et al., 2010), despite
this its formula is employed in WW3 ST4 package. Model results showed that its effects
significantly improve simulations of turbulence characteristics and upper ocean thermal
structure (Huang and Qiao, 2010; Huang et al., 2011). Sea bottom-wave interactions and ice-
wave interactions have been studied in great detail, which are out of scope of present study

and not discussed here.

The objectives of this paper are to explore the dissipation effects of wave-generated
turbulence reacting on ocean waves, and to investigate the role of wave-breaking dissipation
via an improved postbreaking spectrum expression based on the breaking wave statistical
method. The remainder of this paper is organized as follows: Section 2 describes the

analytical approaches for wave energy dissipation induced by wave-generated turbulence and
by random wave-breaking, introduces scale detection comparing to wind input and provides
verifications with laboratory observations or comprehensive measurements; Section 3
presents application implementations on simple duration-limited growth and decay
experiments; Section 4 addresses discussions and issues which need complex insights, and

some conclusions and suggestions for future research are summarized in Section 5.

## 2  Model derivation and verification



## 2.1 The analytical approach for wave energy dissipation induced by wave-generated turbulence

In the usual notation, let $x_i, i = 1, 2, 3$ be rectangular co-ordinates. Let $u_{\mathrm{SM}i}, i = 1, 2, 3$ denote the wave velocities; $T_{\mathrm{SM}}, s_{\mathrm{SM}}, p_{\mathrm{SM}}, \rho_{\mathrm{SM}}$ the perturbations of temperature, salinity, pressure and density induced by ocean waves. More comprehensive governing equations for wave motion were derived in Yuan et al.(2012) and Yang et al. (2019) and the unit volume wave energy balance equation can be obtained in tensor expression as follows:

$$
\begin{aligned}
&\frac{\partial}{\partial t}\left\langle \frac{\rho_0 u_{\mathrm{SM}i}^2}{2} + \frac{g^2 \rho_{\mathrm{SM}}^2}{2\rho_0 \hat{N}_3^2} \right\rangle_{\mathrm{SM}} + \left\langle (\hat{U}_j + u_{\mathrm{SM}j}) \frac{\partial}{\partial x_j} \left( \frac{\rho_0 u_{\mathrm{SM}i}^2}{2} + \frac{g^2 \rho_{\mathrm{SM}}^2}{2\rho_0 \hat{N}_3^2} \right) \right\rangle_{\mathrm{SM}} + \frac{\partial}{\partial x_i} \left\langle p_{\mathrm{SM}} u_{\mathrm{SM}i} \right\rangle_{\mathrm{SM}} \\
&= \left( -\rho_0 \left\langle u_{\mathrm{SM}i} u_{\mathrm{SM}j} \right\rangle_{\mathrm{SM}} \frac{\partial \hat{U}_i}{\partial x_j} + g \left\langle \rho_{\mathrm{SM}} u_{\mathrm{SM}\alpha} \right\rangle_{\mathrm{SM}} \frac{\hat{N}_\alpha^2}{\hat{N}_3^2} \right) + \frac{g^2}{\rho_0 \hat{N}_3^2} \left\langle \rho_{\mathrm{SM}} Q_{\rho \mathrm{SM}} \right\rangle_{\mathrm{SM}} \\
&+ \left\{ 
\begin{aligned}
&\frac{\partial}{\partial x_j} \left\langle \left( \nu_0 + \frac{k^2}{\pi^2 \varepsilon} \right) \frac{\partial}{\partial x_j} \left( \frac{\rho_0 u_{\mathrm{SM}i}^2}{2} \right) \right\rangle_{\mathrm{SM}} + \frac{\partial}{\partial x_j} \left\langle \left( K_0 + \frac{1}{\sigma_0} \frac{k^2}{\pi^2 \varepsilon} \right) \frac{\partial}{\partial x_j} \left( \frac{g^2 \rho_{\mathrm{SM}}^2}{2\rho_0 \hat{N}_3^2} \right) \right\rangle_{\mathrm{SM}} \\
&- \left\langle \rho_0 \left( \nu_0 + \frac{k^2}{\pi^2 \varepsilon} \right) \left( \frac{\partial u_{\mathrm{SM}i}}{\partial x_j} \right)^2 \right\rangle_{\mathrm{SM}} - \left\langle \frac{g^2}{\rho_0 \hat{N}_3^2} \left( K_0 + \frac{1}{\sigma_0} \frac{k^2}{\pi^2 \varepsilon} \right) \left( \frac{\partial \rho_{\mathrm{SM}}}{\partial x_j} \right)^2 \right\rangle_{\mathrm{SM}}
\end{aligned}
\right\}
\end{aligned}
\tag{1}
$$

where $\hat{U}_i, i = 1,2,3; \hat{T}, \hat{s}, \hat{p}, \hat{\rho}$ denote the background current components and $\rho_0$ is the basin mean water density; $\nu_0, \kappa_0, D_0$ the molecular viscosity, thermal and diffusion coefficients;

$\hat{N}_i^2 = -g \dfrac{\partial}{\partial x_i}\left( \dfrac{\hat{\rho}}{\rho_0} \right), i = 1, 2, 3$ the Brunt–Väisälä frequency components; $k^2, \varepsilon$ the kinetic energy

and its dissipation rate of ocean turbulence, which is generated by shear instability of background current (Mellor and Yamada, 1982), Stokes drift (Ardhuin and Jenkins, 2006; Ardhuin et al, 2010; Huang and Qiao, 2010) but mainly generated by ocean waves in the upper layers (Yuan et al., 1999, 2013; Qiao et al., 2004; Yang et al. , 2003, 2004; Babanin, 2006; Babanin and Haus, 2009; Dai et al., 2010; Zhuang et al., 2022). $\frac{\rho_0 u_{\mathrm{SM}i}^2}{2}, \frac{g^2 \rho_{\mathrm{SM}}^2}{2\rho_0 \hat{N}_3^2}$ denote the kinetic and potential wave energy and $\langle \ \cdot \ \rangle_{\mathrm{SM}}$ denotes the Reynolds average on the wave motion. Hereafter, other symbols have their usual meaning. The first term on the left-hand side of Eq. (1) is related to the local mechanical energy variation and the second and third ones denote the energy flux transferred by ocean waves and background currents. The first and second terms on the right-hand side of Eq. (1) are related to the modulation by larger scale motions through shear instability generations. The third term is related to the energy input through thermal radiation, the fourth and fifth ones are related to the modulation by smaller scale motions through ocean mixing and the last two terms are related to the energy





loss rate due to internal viscosity. It should be noted that three types of gravity ocean waves, which consist of surface waves, internal waves and inertial waves, follow the same governing Eq. (1). Here in this study only the former is concerned, so the fifth term on the right-hand side of Eq. (1) is the energy dissipation induced by ocean turbulence (Dissipation induced by

molecular viscosity is insignificant and not considered here).

As stated above, ocean wave-generated turbulence plays a dominant role in the upper layers, the energy loss from ocean waves needs to be studied further. The unit volume energy dissipation mainly induced by wave-generated turbulence can be expressed as

$$-\left\langle \rho_0 \frac{k^2}{\pi^2 \varepsilon} \left( \frac{\partial u_{SMi}}{\partial x_j} \right)^2 \right\rangle_{SM} = -\rho_0 \alpha_{wt} \left\langle \frac{\bar{k}^2}{\pi^2 \bar{\varepsilon}} \right\rangle_{SM} \left\langle \left( \frac{\partial u_{SMi}}{\partial x_j} \right)^2 \right\rangle_{SM} \tag{2}$$

where the symbol overbar '-' denotes the turbulent equilibrium variables, $\left\langle \frac{\bar{k}^2}{\pi^2 \bar{\varepsilon}} \right\rangle_{SM}$ is the so-

called wave-generated turbulent mixing coefficient which was widely used in coupling numerical models. $\alpha_{wt}$ is an undetermined constant here, which implies the quasi-equilibrium level of wave-generated turbulence. Then the total energy dissipation for vertical water column per unit area can be written as

$$-\int_{-\hat{H}}^{0} \left\langle \rho_0 \frac{k^2}{\pi^2 \varepsilon} \left( \frac{\partial u_{SMi}}{\partial x_j} \right)^2 \right\rangle_{SM} dx_3 = -\rho_0 \alpha_{wt} \int_{-\hat{H}}^{0} \left\langle \frac{\bar{k}^2}{\pi^2 \bar{\varepsilon}} \right\rangle_{SM} \left\langle \left( \frac{\partial u_{SMi}}{\partial x_j} \right)^2 \right\rangle_{SM} dx_3 \tag{3}$$

where $\hat{H}$ denotes the water depth.

In the statistical wave theory, the wave field is regarded as weakly-in-the-mean nonlinear processes and the chief linear components are used widely for further detection (Komen et al., 1994). Below we try to derive the total energy dissipation expressed by wavenumber

spectrum through the classical linear wave solutions, i.e.,

$$u_{SM1} = \iint_{\bar{k}} \frac{\omega k_1}{K} A \frac{\cosh K(\hat{H} + x_3)}{\sinh K \hat{H}} \exp\left\{ i \left( \bar{k} \cdot \bar{r} - \omega t \right) \right\} d\bar{k} \tag{4}$$

$$u_{SM2} = \iint_{\bar{k}} \frac{\omega k_2}{K} A \frac{\cosh K(\hat{H} + x_3)}{\sinh K \hat{H}} \exp\left\{ i \left( \bar{k} \cdot \bar{r} - \omega t \right) \right\} d\bar{k} \tag{5}$$

$$u_{SM3} = \iint_{\bar{k}} -\omega i A \frac{\sinh K(\hat{H} + x_3)}{\sinh K \hat{H}} \exp\left\{ i \left( \bar{k} \cdot \bar{r} - \omega t \right) \right\} d\bar{k} \tag{6}$$

As the wave motion is thought to be a stationary and homogeneous stochastic process and

$\left\langle AA^* \right\rangle = \delta(\bar{k} - \bar{k}')E(k_1 k_2)$ where $\delta(\cdot)$ denotes the Dirac function (Kinsman, 2012), then



$$\left\langle \frac{\partial u_{SM1}}{\partial x_1}\frac{\partial u_{SM1}}{\partial x_1}\right\rangle_{SM} = \iint_{\bar{k}} \mathrm{Re}\left\{\iint_{\bar{k}'}\omega\omega'\frac{k_1^2}{K}\frac{k_1'^2}{K'}\frac{\cosh K(\hat{H}+x_3)}{\sinh K\hat{H}}\frac{\cosh K'(\hat{H}+x_3)}{\sinh K'\hat{H}} \atop \left\langle AA^*\right\rangle\exp\left\{i\left[(k_\alpha - k_\alpha')x_\alpha -(\omega-\omega')t\right]\right\}dk_1'dk_2'\right\}dk_1dk_2$$

$$= \iint_{\bar{k}}\mathrm{Re}\left\{\iint_{\bar{k}'}\omega\omega'\frac{k_1^2}{K}\frac{k_1'^2}{K'}\frac{\cosh K(\hat{H}+x_3)}{\sinh K\hat{H}}\frac{\cosh K'(\hat{H}+x_3)}{\sinh K'\hat{H}} \atop \delta(\bar{k}-\bar{k}')E(k_1,k_2)\exp\left\{i\left[(k_\alpha -k_\alpha')x_\alpha-(\omega-\omega')t\right]\right\}dk_1'dk_2'\right\}dk_1dk_2 \qquad (7)$$

$$= \iint_{\bar{k}}\omega^2\frac{k_1^4}{K^2}\frac{\cosh^2 K(\hat{H}+x_3)}{\sinh^2 K\hat{H}}E(k_1,k_2)dk_1dk_2$$

Correspondingly, after some similar manipulations and summing over all product terms, we obtain

$$\left\langle \frac{\partial u_{SMi}}{\partial x_j}\frac{\partial u_{SMi}}{\partial x_j}\right\rangle_{SM} = \iint_{\bar{k}} 2\omega^2 K^2\frac{\cosh 2K(\hat{H}+x_3)}{\sinh^2 K\hat{H}}E(k_1,k_2)dk_1dk_2 \qquad (8)$$

Hence,

$$-\int_{-\hat{H}}^0\left\langle\rho_0\frac{k^2}{\pi^2\varepsilon}\left(\frac{\partial u_{SMi}}{\partial x_j}\right)^2\right\rangle_{SM}dx_3 = -\rho_0 g\iint_{\bar{k}}2\alpha_{wt}\frac{\omega^2 K^2}{g}\left[\int_{-\hat{H}}^0\left\langle\frac{\bar{k}^2}{\pi^2\bar{\varepsilon}}\right\rangle_{SM}\frac{\cosh 2K(\hat{H}+x_3)}{\sinh^2 K\hat{H}}dx_3\right]E(k_1,k_2)dk_1dk_2$$

$$= -\rho_0 g\iint_{\bar{k}}2\alpha_{wt}\frac{\omega^2 K^2}{g}\left\{\frac{\iint_{\bar{k}}\omega^2 K^2\left[\int_{-\hat{H}}^0\left\langle\frac{\bar{k}^2}{\pi^2\bar{\varepsilon}}\right\rangle_{SM}\frac{\cosh 2K(\hat{H}+x_3)}{\sinh^2 K\hat{H}}dx_3\right]E(k_1,k_2)dk_1dk_2}{\iint_{\bar{k}}\omega^2 K^2 E(k_1,k_2)dk_1dk_2}\right\}E(k_1,k_2)dk_1dk_2 \qquad (9)$$

The dissipation source function induced by wave-generated turbulence can be expressed in wavenumber space as

$$S_{tid} = -2\alpha_{wt}\frac{\omega^2 K^2}{g}\left\{\frac{\iint_{\bar{k}}\omega^2 K^2\left[\int_{-\hat{H}}^0\left\langle\frac{\bar{k}^2}{\pi^2\bar{\varepsilon}}\right\rangle_{SM}\frac{\cosh 2K(\hat{H}+x_3)}{\sinh^2 K\hat{H}}dx_3\right]E(k_1,k_2)dk_1dk_2}{\iint_{\bar{k}}\omega^2 K^2 E(k_1,k_2)dk_1dk_2}\right\}E(k_1,k_2) \qquad (10)$$

For deep water depth, it is easily derived as

$$S_{tid} = -4\alpha_{wt}K^3\left\{\frac{\iint_{\bar{k}}\omega^2 K^2\left[\int_{-\infty}^0\left\langle\frac{\bar{k}^2}{\pi^2\bar{\varepsilon}}\right\rangle_{SM}\exp\{2Kx_3\}dx_3\right]E(k_1,k_2)dk_1dk_2}{\iint_{\bar{k}}\omega^2 K^2 E(k_1,k_2)dk_1dk_2}\right\}E(k_1,k_2) \qquad (11)$$

Different from Yuan et al. (2013), in which the power function relationship between turbulent dissipation rate and shear instability generation of wave motion was fitted by observation data in deep ocean, we choose a generic representation of the mixing length of

$\bar{l}_D = \dfrac{\bar{k}^{3/2}}{\pi^{3/2}\bar{\varepsilon}}$ (Baumert et al., 2005), which is appropriate for deep and shallow water conditions.



Then the mixing coefficient $\left\langle \dfrac{\overline{k}^2}{\pi^2 \overline{\varepsilon}} \right\rangle_{\mathrm{SM}}$ is formulated conveniently as

$$\left\langle \frac{\overline{k}^2}{\pi^2 \overline{\varepsilon}} \right\rangle_{\mathrm{SM}} = \frac{\sqrt{7}}{2} \iint\limits_{\hat{k}} E(k_1,k_2) \frac{\cosh^2 K(\hat{H}+x_3)}{\sinh^2 K\hat{H}} dk_1 dk_2 \left( \iint\limits_{\hat{k}} \omega^2 K^2 \frac{\cosh 2K(\hat{H}+x_3)}{\sinh^2 K\hat{H}} E(k_1,k_2) dk_1 dk_2 \right)^{\!\!1/2} \tag{12}$$

For deep water depth, Eq. (12) is reduced to

$$\left\langle \frac{\overline{k}^2}{\pi^2 \overline{\varepsilon}} \right\rangle_{\mathrm{SM}} = \sqrt{\frac{7}{2}} \iint\limits_{\hat{k}} E(k_1,k_2) \exp\{2Kx_3\} dk_1 dk_2 \left( \iint\limits_{\hat{k}} \omega^2 K^2 E(k_1,k_2) \exp\{2Kx_3\} dk_1 dk_2 \right)^{\!\!1/2} \tag{13}$$

Now we try to present a concise description of Eq. (11) for future practical application, the unified mean $\hat{\omega}, \hat{K}$ are introduced for various integral mean variables and consequently, the excessive and tedious mathematical deduction is not involved. Then Eq. (13) is reduced to

$$\left\langle \frac{\overline{k}^2}{\pi^2 \overline{\varepsilon}} \right\rangle_{\mathrm{SM}} \approx \sqrt{\frac{7}{2}} \hat{\omega}\hat{K} \left( \iint\limits_{\hat{k}} E(k_1,k_2) dk_1 dk_2 \right)^{\!\!3/2} \exp\{3\hat{K}x_3\} \tag{14}$$

By employing Eq. (14), Eq. (11) is derived approximately as

$$S_{\mathrm{tid}} \approx -\frac{2\sqrt{14}}{5} \alpha_{\mathrm{wt}} K^3 \hat{\omega} \left( \iint\limits_{\hat{k}} E(k_1,k_2) dk_1 dk_2 \right)^{\!\!3/2} E(k_1,k_2) \tag{15}$$

For finite water depth, a simple scaling factor $R_H = 1 - \exp(-5\hat{k}_H \hat{H})$ is introduced to Eq. (15) for numerical implementation.

We further detect the scales of the dissipation rate comparing to the wind input source function. There are still a relative large uncertainties remained in different bulk energy input
approaches, here only the parameterizations of growth rate due to wind, which were proposed by Komen et al. (1984) and Janssen (1991), are concerned for the following convenient arguments. The growth rate of the wave scales with wavenumber was thought of as $\gamma \sim k^{3/2}$ (Janssen 1991), for $\gamma \sim \beta \dfrac{\omega}{C^2} u_*^2 = g^{-1/2} \beta K^{3/2} u_*^2$, where $\beta$ denotes the Miles parameter $\beta = \dfrac{1.2}{\kappa^2} \mu \ln^4 \mu, \; \mu \leq 1$ and $\mu$, the dimensionless critical height. Moreover, it can also be
rewritten as

$$\gamma \sim \beta \frac{\omega}{C^2} u_*^2 = K\beta \left( \frac{u_*}{C} \right) u_* = K\beta\alpha^{-1/2}(Kz_0)^{1/2} u_* = K\tilde{l}_J u_*, \quad \tilde{l}_J = \alpha^{-1/2} \beta (Kz_0)^{1/2} \tag{16}$$

where $z_0$ is the sea surface roughness, and $\alpha$, the Charnock constant. The dimensionless



variable $\tilde{l}_J \sim \beta (Kz_0)^{1/2}$ is related to the dimensionless critical height $\mu$ and the relative

roughness length $Kz_0$. Incidentally, the growth rate parameterized by Komen et al. (1984) is

fairly linearized as follows:

$$\gamma' \sim \frac{\omega}{C} u_* = K u_* .$$

From Eq. (15), the dissipation rate induced by wave-generated turbulence can be rescaled

as

$$
\begin{aligned}
\gamma_{\text{tid}} &\sim \frac{2\sqrt{14}}{5} \alpha_{\text{wt}} K^3 \hat{\omega} \frac{A^3}{2^{3/2}} \\
&= \frac{\sqrt{7}}{5} \alpha_{\text{wt}} K^3 A^2 (A\hat{\omega}) = \frac{\sqrt{7}}{5} \alpha_{\text{wt}} K (KA)^2 u_{\text{w0}} = K\tilde{l}_Y u_{\text{w0}}, \qquad \tilde{l}_Y = \frac{\sqrt{7}}{5} \alpha_{\text{wt}} (KA)^2
\end{aligned}
\tag{17}
$$

where $A$ is the wave amplitude, and $u_{\text{w0}}$, the wave orbital velocity at sea surface. $\tilde{l}_Y \sim (KA)^2$

can be taken to be some kind of dimensionless height or wave steepness. Hence Eq. (16) and

(17) yield a considerable uniformity of analytical expressions. Figure 1 shows the growth rate

$\gamma$ roughly calculated under the condition $\frac{\tau_{\text{w}}}{\tau} = 0.5$ for simplicity, and Fig. 2 shows the

dissipation rate $\gamma_{\text{tid}}$ with $\alpha_{\text{wt}} = 1.0$. Both are comparable in spatial distribution and magnitude,

especially under normal and extreme sea conditions. And the corresponding spectral

signatures of difference between Fig. 1 and 2 dominate the wave growth or decay.

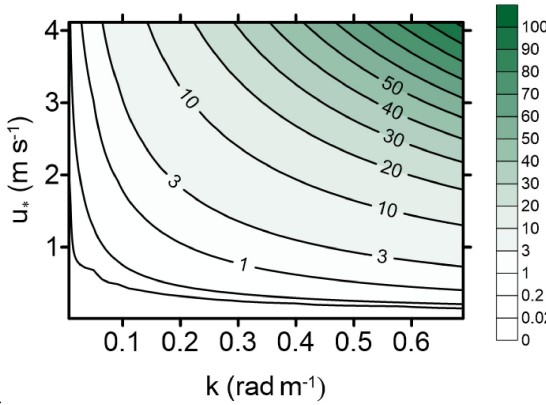

**Figure 1**. Distribution of growth rate $\gamma$ as a function of wavenumber and friction
velocity (Unit: s⁻¹)



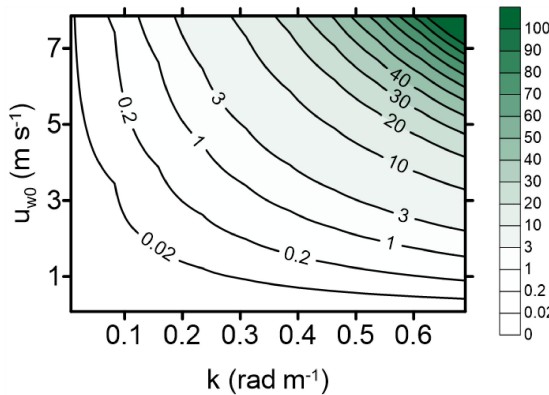

**Figure 2**. Distribution of dissipation rate $\gamma_{tid}$ as a function of wavenumber and wave orbital velocity at sea surface (Unit: s$^{-1}$)

The unit volume wave energy dissipation rate induced by wave-generated turbulence is

$$e_{tid} = \left\langle \frac{k^2}{\pi^2 \varepsilon} \left( \frac{\partial u_{SMi}}{\partial x_j} \right)^2 \right\rangle_{SM} = \alpha_{wt} \left\langle \frac{\overline{k}^2}{\pi^2 \overline{\varepsilon}} \right\rangle_{SM} \left\langle \left( \frac{\partial u_{SMi}}{\partial x_j} \right)^2 \right\rangle_{SM}$$ following from Eq. (2), and according the

equilibrium solutions of the generation term to the dissipation for wave-generated turbulence (Yuan et al., 2013), the TKE dissipation rate $\varepsilon_{dis}$ can be virtually identical to $e_{tid}$, i. e.,

$$\varepsilon_{dis} \approx e_{tid} = \alpha_{wt} \left\langle \frac{\overline{k}^2}{\pi^2 \overline{\varepsilon}} \right\rangle_{SM} \left\langle \left( \frac{\partial u_{SMi}}{\partial x_j} \right)^2 \right\rangle_{SM} \tag{18}$$

There are many studies to verify the modeled TKE dissipation rate $\varepsilon_{dis}$, generated by shear instability of irregular wind waves or swells, with cruise observations (Yuan et al., 2013; Zhuang et al., 2020, 2021). Here a direct comparison between modeled $\varepsilon_{dis}$ with laboratory observations is performed for monochromatic non-breaking waves, under which circumstances Eq. (18) can be easily derived as

$$\varepsilon_{dis}(x_3) = \frac{7\sqrt{7}}{16} \alpha_{wt} A^5 \omega^3 K^3 \exp\{5Kx_3\} \tag{19}$$

where $A$ is the monochromatic wave amplitude at surface. The ''observed'' TKE dissipation rates $\varepsilon_{dis}$ come from the laboratory measurements conducted in the Air-Sea Interaction Saltwater Tank (ASIST) of the University of Miami (Babanin and Haus, 2009; Babanin, 2011), with generated wave trains of $f$=1.5 Hz and $K_{1.5Hz}$= 9.82 rad m$^{-1}$. The significant TKE dissipation rates $\varepsilon_{dis}$ were retrieved from the wavenumber spectra provided by Particle Image Velocimetry (PIV) measurements at the 30 mm layer from the still surface. Their detailed





experimental measurements indicate that the turbulence observed must have been directly

generated by the waves themselves (Babanin and Haus, 2009). Figure 3 shows the

dependence of TKE dissipation rate $\varepsilon_{dis}$ versus wave amplitude $A$ in logarithmic and linear

scales respectively, and the solid line is plotted by using Eq. (19) with $\alpha_{wt} = 1.0$ ( Layer depth

$x_3$ varies corresponding to different $A$). Babanin (2011) interpreted that the ''observed'' $\varepsilon_{dis}$

are instantaneous values incurred intermittently at the rear-face phase of the wave below the

level of the wave trough, and if averaged over the wave period, the estimates of $\varepsilon_{dis}$ have to

be divided at least by a factor of 10 and perhaps more. This implies that such intermittent

turbulence is still at the stage of quasi-equilibrium level, and the coefficient $\alpha_{wt} = 1.0$ should

be tuned to less than one order of magnitude or more for practical application (Wang et al.,

280 2024).

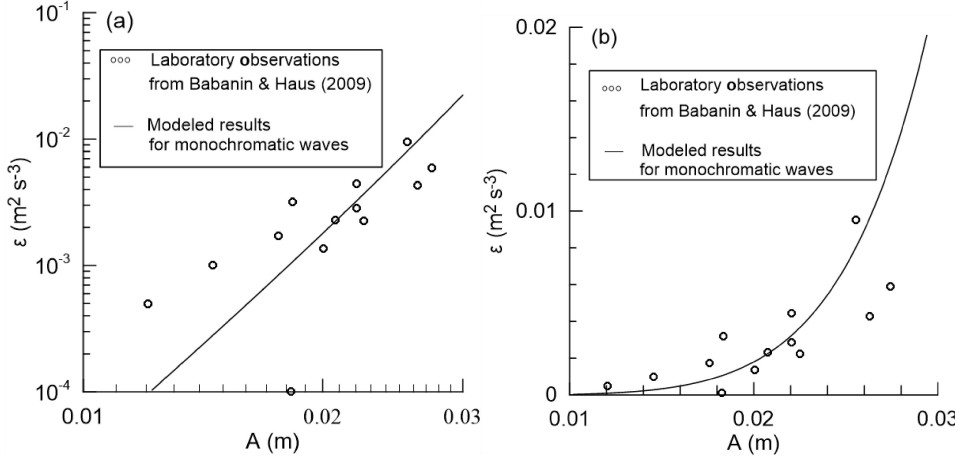

**Figure 3**. Dependence of TKE dissipation rate $\varepsilon_{dis}$ (denoted as $\varepsilon$ in the figure) versus
wave amplitude $A$. Observation data (circles) are digitalized from Babanin and Haus (2009).
Dependence (19) is shown with a solid line. Data are plotted in **(a)** logarithmic scales and **(b)**
linear scales.

The following ''observed'' TKE dissipation rates $\varepsilon_{dis}$ with different layer depth come

from the laboratory measurements performed inside a wave tank at the Institute of Applied

Physics of the Russian Academy of Science, Nizhny Novgorod, Russia (Wei et al., 2018).

The tank was equipped with a mechanical wavemaker and a fan, which are used to generate

mechanical and wind waves respectively. The vertical surface displacement of the generated

wind waves and swell trains were measured by 3 resistancetype wave gages, and series of

larger waves are selected for our further comparisons (Table 1).





| Experiment number | Wave condition | $H$s (cm) | Peak frequency (Hz) | Wave length (m) |
|---|---|---|---|---|
| 1 | 21 m s$^{-1}$ wind | 3.75 | 2.72 | 0.42 |
| 2 | 32 m s$^{-1}$ wind | 6.03 | 1.97 | 0.83 |
| 3 | 1.6 Hz swell with 21 m s$^{-1}$ wind | 5.26 | 1.64 | 0.86 |
| 4 | 1.2 Hz swell | 3.98 | 1.20 | 1.36 |

**Table 1**. Significant wave height ($H$s) and peak frequency for selected wave conditions

The TKE dissipation rates $\varepsilon_{dis}$ at different layers from surface were retrieved from the wavenumber spectra provided by underwater 3-D instantaneous velocity measured by using an Acoustic Doppler Velocimeter (SonTek microADV). As Thais and Magnaudet (1996) interpreted their experimental observations, the wave orbital motions, which possess strong
vertical gradients, ought to be the dominate role of enhancing of the turbulence production. Figure 4 shows the dependence of TKE dissipation rate $\varepsilon_{dis}$ versus layer depth in linear/logarithmic scale for various wave conditions respectively, and the solid lines are plotted also by using Eq. (19) with $\alpha_{wt} = 1.0$ ( $A$ is the mean amplitude of the highest one-third or one-half waves). The decreasing tendency of modeled $\varepsilon_{dis}$ with layer depth under different
wave conditions agrees with that of observations in the upper 0.2 m layers. It should be noted that when the layer depth is larger than 0.25 m, the ''observed'' $\varepsilon_{dis}$ may be regarded as adaptive noises induced by other dynamic mechanisms.



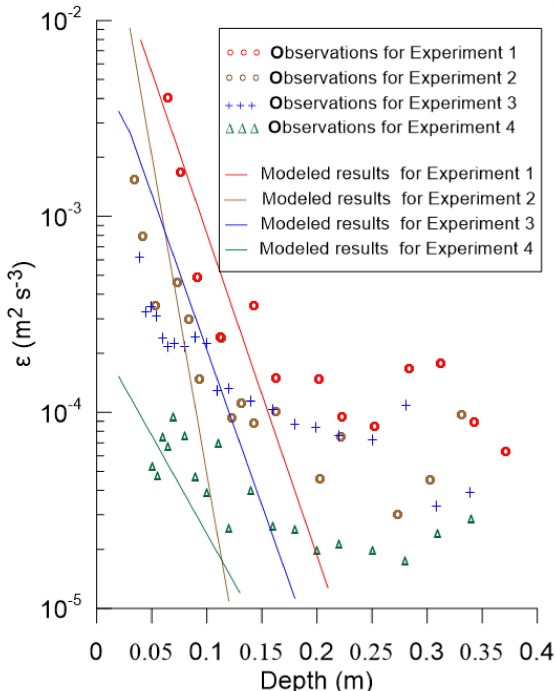

**Figure 4**. Dependence of TKE dissipation rate $\varepsilon_{dis}$ (denoted as $\varepsilon$ in the figure) versus layer depth. Observation data (circles, pluses and triangles) are digitalized from Wei et al. (2018). Dependence (19) is shown with solid lines. Data are plotted in linear/logarithmic scale on the horizontal/vertical axis.

## 2.2 The precise estimation of wave energy loss induced by random wave-breaking

Wave-breaking is another critical dissipation process of ocean waves, which is highly intermittent but simultaneous with wave-generated turbulence stated above. However, the former mechanism still remains not well-understood (Donelan and Yuan, 1994; Young and Babanin, 2006; Yuan et al., 2009). Here in this study, based on the breaking wave statistical method that Cartwright and Longuet-Higgins (1956), Longuet-Higgins (1957), Yuan et al (1986, 2009) employed, we present a more precise approach for estimation of the dissipation-due-to-breaking. Parameterization treatments of the dissipation source function under the assumption of balance between growth and dissipation are discarded, instead of which we derive an improved analytical postbreaking spectrum expression satisfying the kinematic and dynamic wave-breaking onset criterions.





The postbreaking wave spectrum, via the covariance of surface elevation which was assumed to be Gaussian and stationary, was expressed as (Yuan et al., 1986;Yuan et al.,1993; Donelan and Yuan, 1994)

$$E_b(k_1,k_2) = \alpha_b E(k_1,k_2) \tag{20}$$

and

$$\alpha_b = [1-(\frac{\omega}{\omega_b})^2(\frac{2}{\pi})^{1/2}\frac{1}{L}\exp(-\frac{L^2}{2})]^2 \tag{21}$$

where $L^{-1} = \dfrac{\mu_4^{1/2}}{g/2}$ denotes the normalized *rms* acceleration, $T_b = 2\pi(\frac{\mu_2}{\mu_4})^{1/2} = 2\pi(\frac{\mu_0}{\mu_2})^{1/2}$ $\rho = \rho T_z$

and $\omega_b = \dfrac{2\pi}{T_b}$ denote the mean period or mean frequency of wave maxima, $T_z$ denotes the zero-crossing wave period; $\rho$ denotes a parameter associated with the spectrum width

parameter $\varepsilon$, i.e.,

$$\rho^2 = \frac{\mu_2^2}{\mu_0\mu_4} \tag{22}$$

$$\varepsilon^2 = 1-\rho^2 \tag{23}$$

and $\mu_i$, the *i-th* order moment of the wave spectrum. It should be noted that all the variables stated above are related to the incipient waves, not the postbreaking ones.

So the attenuation coefficient $\alpha_b$ is derived as

$$\alpha_b = \left[1-4(2\pi)^{-1/2}\frac{\rho\omega^2\mu_0^{1/2}}{g}\exp(-\frac{\rho^2}{8}\frac{g^2}{\mu_0\omega_z^4})\right]^2 \tag{24}$$

and the ratio of total energy loss due to wave-breaking is given by

$$r_b = \frac{\iint_k E(k_1,k_2)dk_1dk_2 - \iint_k E_b(k_1,k_2)dk_1dk_2}{\iint_k E(k_1,k_2)dk_1dk_2} \approx 8(2\pi)^{-1/2}\frac{\rho\bar\omega^2\mu_0^{1/2}}{g}\exp(-\frac{\rho^2}{8}\frac{g^2}{\mu_0\omega_z^4}) \tag{25}$$

where $\bar\omega$ denotes the integral mean wave frequency. Babanin (2006) analyzed the

measurement records by Yefimov and Khristoforov (1971) and concluded that the breaking ratio of dominant waves was 0.01-0.4 %.



In the neighborhood of wave crests $\mu_0\bar{\omega}^2 \sim \frac{1}{2}c_0^2$, so

$$\frac{\bar{\omega}^2\mu_0^{1/2}}{g} = \frac{\bar{\omega}^2\mu_0}{g\mu_0^{1/2}} \sim \frac{c_0^2}{2g\mu_0^{1/2}} = \frac{g\mu_2}{2\mu_0^{1/2}\mu_4} = \rho\frac{g}{2\mu_4^{1/2}} = \rho L.$$ This yields $r_b \sim \rho^2 L$, which agrees with the

relative mechanical energy loss per unit sea surface area. Wang et al. (2017, 2018) concluded

that the ratio of the kinetic energy loss to potential one is mainly within the range 3-30, which

indicates that there is a disproportion feature between the wave kinetic energy loss and

potential one due to wave-breaking. Shi et al. (2025) validated the statistical wave-breaking

model across multiple sites from the High Wind Speed Gas Exchange Study (HiWinGS), and

concluded that the model is highly effective in capturing the dynamics of whitecap coverage

across a range of high sea states. Hence we propose an improved attenuation coefficient

expressed as follows:

$$\alpha_b' = 1 - \rho\frac{\omega^2\mu_0^{1/2}}{g}\frac{1}{L}\left[L\int_{-\infty}^{-L}\exp\left\{-\frac{1}{2}Z^2\right\}dZ + \exp\left\{-\frac{\rho^2}{8}\frac{g^2}{\mu_0\omega_z^4}\right\}\right] \qquad (26)$$

where $L = \frac{g}{2\mu_4^{1/2}} = \frac{\rho}{\pi\lambda}\left(\frac{H_s}{\bar{L}}\right)^{-1} = \frac{\rho}{2}\frac{g}{\mu_0^{1/2}\omega_z^2}$, mean wavelength $\bar{L} = \lambda\frac{g}{2\pi}T_z^2$, $\lambda = \frac{2}{3}$ or 0.86 (Kinsman,

2012; Yuan et al., 2009; Xu and Yu, 2001). There is a prominent consistency between Eq.

(26) and Eq. (24) under some certain circumstances. Let

$$\theta \equiv L\int_{-\infty}^{-L}\exp\left\{-\frac{1}{2}Z^2\right\}dZ \Big/ \exp\left\{-\frac{\rho^2}{8}\frac{g^2}{\mu_0\omega_z^4}\right\}$$ represents the ratio of the kinetic energy loss to

the potential one due to wave-breaking (Yuan et al., 2009; Wang et al., 2017, 2018) , Eq. (26)

can be rewritten as

$$\alpha_b' = 1 - \rho\frac{\omega^2\mu_0^{1/2}}{g}\frac{1+\theta}{L}\exp\left\{-\frac{\rho^2}{8}\frac{g^2}{\mu_0\omega_z^4}\right\} \qquad (27)$$

Suppose that $\frac{H_s}{\bar{L}} = \frac{1}{7}, \rho = 0.6, \theta = 4$, then $\frac{1+\theta}{L} \doteq 3.216 \sim 8(2\pi)^{-1/2} \doteq 3.192$, while the

latter coefficient comes originally from the complicated 0-1$^{st}$ order asymptotic expansions of

the covariance of surface elevation. Equations (26) and (27) demonstrate definitely the

dominant role of the kinetic energy loss induced by wave-breaking, and the former is applied

in the following numerical experiments.

Young and Babanin (2006) obtained the breaking spectrum and nonbreaking spectrum

from segments of the comprehensive measurement records in the Australian Shallow Water





Experiment (AUSWEX), carried out at Lake George in New South Wales in 1997-2000, and analyzed the spectral difference with the ratio of the two spectra plotted as a function of frequency $f$ in Fig. 5. According to Eq. (26), we also calculate the attenuation coefficient with

$H$s = 0.45 m, $f_\mathrm{p}$ = 0.39 Hz but $\rho$ differs from 0.4 to 0.6. Although there is oscillation mainly caused by survey array of high-precision capacitance wave probes, a decreasing tendency of the ''observed'' ratio is remarkable from low frequencies to the high frequency $f = 5f_\mathrm{p}$, and the calculated attenuation coefficients correspond to this tendency much. But for the higher frequency $f > 5f_\mathrm{p}$, there is not the decreasing feature for the ''observed'' ratio, this can be

explained by the statistical equilibrium in the equilibrium range proposed by Phillips (1985) or the short scales' prompt recovery interpreted by Young and Babanin (2006). When Eq. (26) is applied in the 3$^{rd}$ generation wave models, e.g. the MASNUM wave model (Yuan et al, 1991; Yang et al., 2005), though the attenuation coefficient $\alpha_\mathrm{b}^!$ approaches to zero for those higher frequencies, the equilibrium wave spectra in the equilibrium range are used to

complement the underestimation induced by wave-breaking.

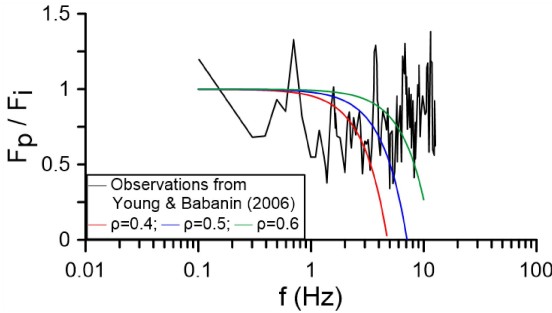

**Figure 5**. Ratio of the spectra between incipient-breaking and postbreaking waves. Black line: Observations are digitalized from Young and Babanin (2006); Color lines: Calculated $\alpha_\mathrm{b}^!$ according to Eq. (26) with $\rho$=0.4, 0.5, 0.6 respectively.

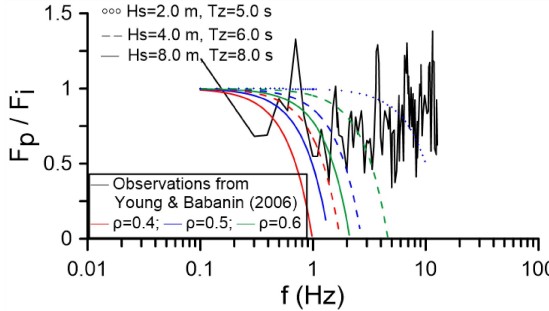




**Figure 6**. Same as Fig. 5 but for ordinary and high sea states. Observations digitalized from Young and Babanin (2006) are also displayed for comparison. Circles, dashed and solid lines represent different wave states, while colors represent different $\rho$ selected in Eq. (26).

We further evaluate the attenuation coefficients by using Eq. (26) on ordinary and high sea states, which are plotted in Fig. 6. Apart from the significant wave height, the zero-crossing wave period also plays an important role for the quantity of the attenuation coefficient. In addition, it decreases apparently at high sea states, which indicates the wave-breaking intensity is more remarkable.

## 3 Application implementations on simple duration-limited growth and decay experiments

There remain two undetermined parameters $\alpha_{wt}$, $\rho$ in the preceding section, which will be discussed further in Section 4. Both of them are chosen as tunable parameters in the duration-limited growth and decay experiments undertook below. The numerical experiments were carried out with the MASNUM wave model (Yuan et al, 1991; Yang et al., 2005), and implemented according to the studies of Janssen et al. (1994). We ran the model for seven days, the first two days with a wind speed of 18.45 m s⁻¹, after which the wind dropped to a value of 5 m s⁻¹. The model integration time step was 30 s, so the growth limiter can be switched off and its impacts need not be considered here.

For simplicity, the parameterization of linear growth in spectral density proposed by Snyder et al. (1981) and Komen et al. (1984) and the quasi-linear one by Janssen (1991) for wind input source function $S_{in}$ are used respectively. As stated above, coefficient $\alpha_{wt}$ should be tuned to 0.02 - 0.2. Wang et al. (2017, 2018) validated the sea surface whitecap courage obtained from the statistical wave-breaking model with the satellite-derived data and proposed that the range of $\rho$ is 0.53 - 0.59, and this referenced span is used in the following experiments.

To validate the model effects proposed in Section 2, several numerical experiments are carried out (Table 2). In the original 3ʳᵈ generation MASNUM wave model described here (Experiment 1), a parameterization proposed by Yuan et al. (1986),Yuan et al. (1993), Donelan and Yuan (1994) is adopted for wave-breaking dissipation source function $S_{ds}$, in which two critical coefficients $d_1$ and $d_2$ were retrieved through fitting algorithm with the dimensional expression proposed by Komen et al. (1984). In Experiment 2, instead of this source function $S_{ds}$, Eqs. (20) and (26) are used to calculate the postbreaking spectrum, while





other unchanged source functions $S_{in}$, $S_{nl}$, $S_{bo}$ and $S_{cu}$ are integrated to obtain the incipient-breaking spectrum. Then the dissipation source function induced by wave-generated turbulence $S_{tid}$ is further considered in Experiment 3. In the above Experiments 2 and 3, $\rho = 0.59$ is chosen, while $\rho = 0.53$ in the supplemental Experiment 2S for further comparison.

| Experiment number | Original $S_{ds}$ | Equations (20) and (26) (Improved postbreaking spectrum $E_b(k_1,k_2)$) | Equation (15) (Dissipation source function $S_{tid}$ induced by wave-generated turbulence) |
|---|---|---|---|
| 1 | √ | × | × |
| 2 | × | √ ( $\rho = 0.59$ ) | × |
| 2S | × | √ ( $\rho = 0.53$ ) | × |
| 3 | × | √ ( $\rho = 0.59$ ) | √ |

**Table 2**.Numerical experiments

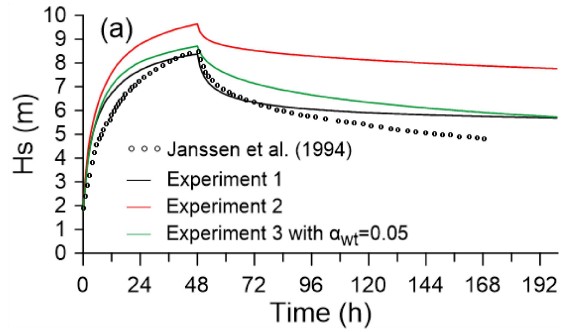

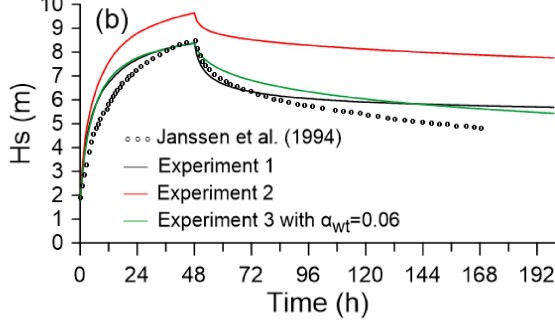

**Figure 7**. Time evolution of wave height over a seven day period. Modeled data (circles) are digitalized from Janssen et al. (1994). After two days the wind drops. Notice the decay in wave height during the last five days when the waves are considered as swell. **(a)** $\alpha_{wt} = 0.05$ ; **(b)** $\alpha_{wt} = 0.06$ .




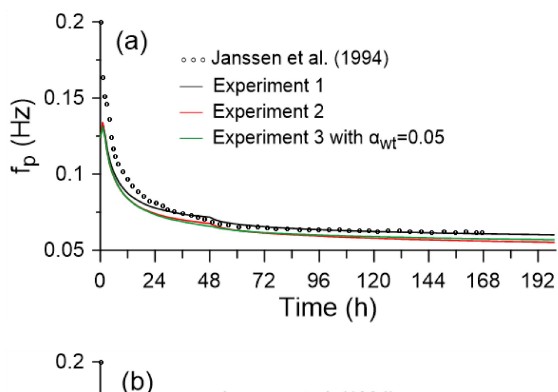


**Figure 8**. Time evolution of peak frequency over a seven day period. Modeled data (circles) are digitalized from Janssen et al. (1994). After two days the wind drops. Notice the decay in the slight shift in peak frequency during the last five days when the waves are
considered as swell. **(a)** $\alpha_{wt} = 0.05$ ; **(b)** $\alpha_{wt} = 0.06$ .

Figures 7 and 8 show the time evolutions of wave height and peak frequency, where the wind input source function in MASNUM wave model was adopted from Snyder et al. (1981) and Komen et al. (1984). The wave height grows during the first two days, then decreases significantly after the wind drops two days later. Results of Janssen et al. (1994) with circle
symbols are listed for comparison with Experiment 1-3. Results of the original MASNUM wave model (Experiment 1) are consistent with that of Janssen et al. (1994), but the deviation of wave height may be noted 96h later due to different power of normalized wave slope in the wave-breaking dissipation source function (Yuan et al., 1991; Donelan and Yuan, 1994). The difference of wave height between Experiment 2 and others, both the maximum quantity and
swell decay, can be distinguished apparently. Especially the wave height in Experiment 2 hardly changes during the swell decay process, because the mean swell steepness becomes small gradually and $\alpha'_b \sim 1.0$ in Eqs. (20) and (26). This indicates that the wave energy loss induced by wave-breaking is inadequate, and the role of prior proposed parameterizations for wave-breaking dissipation may be overestimated in the previous studies. Besides the effect of
postbreaking wave spectrum, the dissipation source function induced by wave-generated





turbulence $S_{tid}$ is incorporated in Experiment 3. The corresponding modeled wave height and

peak frequency are also listed in Figs. 7 and 8, in which different $\alpha_{wt} = 0.05$ or $0.06$ is

selected to highlight its effect. Discrepancies for the maximum of wave height and swell

decay are reduced much than that of Experiment 2, and its variation has an analogous trend

with that of Janssen et al. (1994).

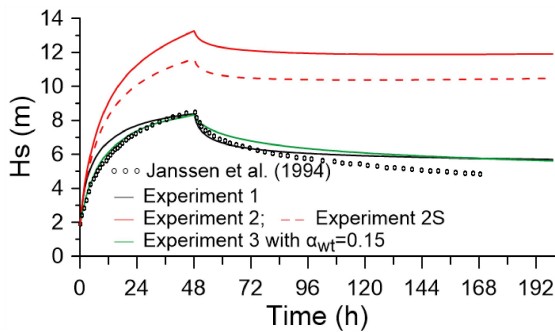

**Figure 9**. Same as Fig. 7 but the wind input source function in MASNUM wave model
was adopted from Janssen (1991) and $\alpha_{wt} = 0.15$.

Figure 9 shows the time evolution of wave height, where the wind input source function

in MASNUM wave model was adopted from Janssen (1991). Similar interpretation stated

above can be obtained, besides that the effect of postbreaking wave spectrum is still

inadequate even for that in the supplemental Experiment 2S where $\rho = 0.53$.

## 4  Discussions

The analytical model and the corresponding numerical results stated above obtain satisfactory

agreements with observations, improve further understandings of wave energy dissipation

due to wave-breaking and wave-generated turbulence, but there remain problems to be

addressed that $\rho$, $\alpha_{wt}$ are chosen as constants. Parameter $\rho$ is associated with the spectrum

width $\varepsilon$. And according to the statistical theory of breaking waves (Cartwright and Longuet-

Higgins, 1956; Longuet-Higgins, 1957; Yuan et al., 1986, 2009), both parameters $\rho$, $\varepsilon$

should be referred to those of incipient-breaking spectrum, which is obtained as an

intermediate variable in our model where all source functions are integrated except for the

energy loss induced by wave-breaking. In fact, the observed or model outputted wave

spectrum would respond to the postbreaking wave spectrum (Donelan and Yuan, 1994). So

the prior-to-breaking parameter $\rho$ or $\varepsilon$ is currently still poorly understood. Moreover,

Lamarre and Melville (1991), Melville et al. (1992) showed that 30% - 50% of energy lost by





breaking waves is expended on entraining bubbles into the water against buoyancy forces, and the residue contributes to the turbulence generation. The interaction mechanism of the breaking-induced turbulence with the progressive waves is also unknown. It should also be mentioned that the whitecap model originally proposed by Hasselmann (1974) is an after-

breaking class model, and its inherent assumptions need experimental verification (Young and Babanin, 2006). The whitecaps are situated on the forward faces of the waves, exert a downward pressure on the upward moving water, but the direct and precise estimates of their negative work on the waves need further studies. Given the intricate interactions state above, in our model proposed here, the choice of $\rho$ as a tuned parameter is tentative for numerical

implementation.

     The constant $\alpha_{\mathrm{wt}}$ is chosen in such a way as a tuned coefficient that it implies the quasi-equilibrium level of wave-generated turbulence, and physically it depends on the normalized shear instability strength of wave motion and further relates to the normalized $3^{\mathrm{rd}}$-order moment of wave spectrum (see also Eqs. 7-8). Therefore it is certainly not constant at any

instant $t$ in real-world scenarios and its reasonable parameterization should be further studied in the future. This idea is quite similar to the wave-amplitude-based Reynolds number proposed by Babanin (2006), its critical wave Reynolds number for wave-induced turbulence can reach down to a lower margin, $Re_{\mathrm{wave}} \sim 1000$, from a series of laboratory experiments (Babanin and Haus, 2009; Dai, et al., 2010). But the appropriate dissipation rate for model

implements cannot be inferred from the wave Reynolds number alone and was approached by experimental means (Babanin, 2011; Zieger et al., 2015; Liu, et al., 2019). The gradient Richardson number $R_{\mathrm{g}}$ may be a more appropriate dependency factor for the coefficient $\alpha_{\mathrm{wt}}$, which needs further perspectives. If the gradient Richardson number $R_{\mathrm{g}}$ is smaller than a critical value $R_{\mathrm{g}}^{\mathrm{c}}$, instability of the flow occurs and the perturbation must be amplified, i.e.,

$$R_{\mathrm{g}} = \frac{\hat{N}_3^2}{\left|\dfrac{\partial u_{\mathrm{SM}\alpha}}{\partial x_3}\right|^2} < R_{\mathrm{g}}^{\mathrm{c}}, \quad \alpha = 1, 2$$

where $\hat{N}_3^2 = -g\dfrac{\partial}{\partial x_3}\left(\dfrac{\hat{\rho}}{\rho_0}\right)$, the Brunt–Väisälä frequency. Below we let $\hat{N}_3 = \mathrm{const} \approx 0.01$ rad s$^{-1}$ in the upper layers and the critical value $R_{\mathrm{g}} \approx 1/2$ (Baumert and Peters, 2000; Baumert et al., 2005). For a monochromatic non-breaking wave $\zeta = \eta\cos(k_\alpha x_\alpha - \omega t)$, which is deemed as a



base motion in the framework of the ocean dynamic system, the instability criterion is
simplified as

$$\hat{N}_3^2 < R_g^c \omega^2 (AK)^2 \exp(2Kx_3)\cos^2(k_\alpha x_\alpha - \omega t)$$

We roughly estimate the critical instability depth induced by the monochromatic waves
(Table 3), Fig. 10 shows the dependence of the gradient Richardson number versus layer
depth for Case 1 in Table 3. Below the wave crests and troughs, there exist bowl-shaped
instability regions, which agree to the experimentally observed instantaneous turbulence
incurred intermittently at the rear face of the progressive wave profile, but the breaking-in-
progress turbulence develops at the front face (Babanin and Haus, 2009; Babanin, 2011).

| No. | Circular frequency $\omega$ (rad s$^{-1}$) | Wave steepness $AK$ | Wavenumber $K$ (rad m$^{-1}$) | Critical instability depth $x_3^c$ (m) |
|---|---|---|---|---|
| 1 | 1.0 | 0.1 | 0.1 | 19.6 |
| 2 | 1.0 | 0.3 | 0.1 | 30.5 |
| 3 | 0.63 | 0.1 | 0.04 | 37.2 |
| 4 | 0.63 | 0.05 | 0.04 | 19.9 |

**Table 3**. Critical instability depth induced by the monochromatic waves

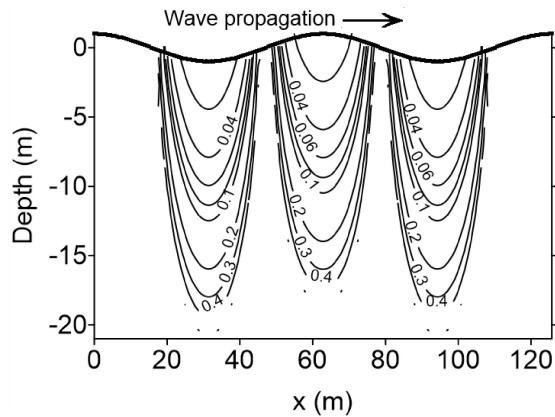


**Figure 10**. Dependence of the gradient Richardson number versus layer depth for

Case 1 in Table 3.



## 5 Conclusions

The ocean wave energy dissipation is the least understood of the major source terms, especially for the energy loss due to wave-breaking and wave-turbulence interaction. Previous attempts, either through experimental or through analytical (or both) approaches as described in section 1, were too ''excessively'' parameterized and resulted in their physical models being much vulnerable.

In the present paper, we attempted to explore the dissipation effects of wave-generated turbulence reacting on ocean waves, and to estimate the energy loss more precisely due to wave-breaking via an improved postbreaking spectrum expression based on the breaking wave statistical method. Theoretical investigations and numerical experiment discussions in sections 2 and 3 argue that these analytical approaches should be able to provide more 535 reliable estimates of the above two dissipation processes.

The main conclusion of the study is that we propose an analytical dissipation source function induced by wave-generated turbulence $S_{\mathrm{tid}}$ formulated by Eq. (15), together with an improved postbreaking spectrum expression $E_{\mathrm{b}}(k_1,k_2)$ by Eqs. (20) and (26). The former dissipation term represents definitely the feedback of imparting of wave shear instability 540 generations on turbulence, and the latter expression depicts the intermittent wave-breaking events more explicitly. Sum of both contributions therefore play critical role of wave energy dissipation.

Another important conclusion of the paper is that the wave energy loss only induced by wave-breaking is much inadequate, which indicates that it may be somewhat overestimated in 545 the previous studies. For the after-breaking approach stated in the whitecap model, the remote-sensing techniques provide a potential way for estimating precisely its downward pressure on the upward moving water even negative work on the waves through satellite retrieved wave-breaking parameters such as whitecap coverage, foam-layer thickness, etc, which needs further elaborations.

Here in this paper we roughly estimate the effects of the above two dissipation mechanisms in simple duration-limited growth and decay experiments. Calibration and verification against a series of academic and realistic simulations, including the fetch/duration-limited cases, turning wind (e.g. cold waves or monsoon) / rotatory wind (e.g.



extratropical or tropical cyclone) conditions, numerical hindcast and operational forecast in
regional and global oceans, will be pursued in our future project, together with considering
other concerned dissipation mechanisms.

*Code and data availability.* Code and *d*ata are available from the authors upon reasonable
request.


*Author contributions.* **Conceptualization**, YongzengYang, Xunqiang Yin.; **Methodology**,
Yongzeng Yang, Fuwei Wang, Xingjie Jiang; **Software**, Fuwei Wang, Xunqiang Yin,
Yongzeng Yang; **Validation**, Yongzeng Yang, Fuwei Wang, Xingjie Jiang; **Formal analysis**,
Fuwei Wang, Yongzeng Yang, Meng Sun; **Investigation**, Meng Sun, Yongzeng Yang;
**Resources**, Meng Sun, Yongfang Shi; **Data curation**, Yongfang Shi, Meng Sun; **Writing—
original draft preparation**, Yongzeng Yang, Meng Sun; **Writing—review and editing**,
Fuwei Wang, Yong Teng, Xunqiang Yin, Yongfang Shi; **Visualization**, Meng Sun,
Yongfang Shi; **Supervision**, Xingjie Jiang, Xunqiang Yin; **Project administration**,
Yongfang Shi.; **Funding acquisition**, Meng Sun,Yongzeng Yang.


*Competing interests.* The authors declare that they have no conflict of interest.

*Acknowledgments.* We especially thank distinguished supervisors for their opinion and
expertise given in the seminar series of the Ocean Dynamic System Team in the MASNUM
Lab.(Yuan et al., 2012, 2013). We also appreciate the comprehensive WW3DG user manual
and useful  WAVEWATCH III software packages (WW3DG, 2019) for reference to improve
our operational FORTRAN source codes.

*Financial support.*  This research was jointly supported by the National Key Research
Program of China, under grant nos. 2023YFC3008200 and 2022YFC3104800; The National
Program on Global Change and Air-Sea Interaction (Phase II), under grant no. GASI-04-
WLHY-02 and Laoshan Laboratory Fund, under grant no. LSKJ202203003.

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
