# Peer review of "Analytical approaches for wave energy dissipation induced by wavegenerated turbulence and random wave-breaking"

_EGUsphere, 2025_

## Referee Comment (RC2)

**Review report**

Preprint egusphere-2025-2671

Title: Analytical approaches for wave energy dissipation induced by wave-generated turbulence and random wave-breaking

Authors: Yongzeng Yang, Fuwei Wang, Meng Sun, Xingjie Jiang, Xunqiang Yin, Yongfang Shi, and Yong Teng

**Overall assessment**

This study proposes revised parameterizations for wave dissipation from two aspects: wave-generated turbulence and wave-breaking. Through the analysis of mechanical energy equation for wave orbital motion, the authors present a revised formulation of wave attenuation through wave-turbulence interaction, which was empirically fit to observation data previously. As for the wave-breaking induced attenuation, they employ a formulation such that the spectrum adjusts to a prescribed post-breaking spectrum. They also investigate the general behavior of the new formulations through some idealized simulations.

The topic of wave dissipation is certainly important and falls within the scope of Ocean Science. However, the theoretical and experimental development in its current form is not convincing enough. The derivation lacks clear statements of assumptions and several steps in the derivation lack sufficient justification, both mathematically and physically. Moreover, the numerical tests are conducted without observational or analytical benchmarks, leaving their purpose and insights unclear. These issues make it difficult to assess the validity and appropriateness of the proposed formulation, which is central to the manuscript.

The manuscript would benefit from language editing by either a native English speaker or a professional editor.

**Major comments**

1.  Abstract: Given the insufficient theoretical and experimental foundations as pointed out in the comments below, the abstract appears to overstate the fidelity of the findings of the present study. For example:
    - L6: The presented formulation cannot be described as "improved" without rigorous theoretical foundation or adequate comparison against existing models.
    - L8: The comparison with measurements or previous simulation results and associated discussions are too limited and do not adequately "verify" the new formulation.

2. The analytical derivation processes in Section 2 lack clarity in many parts. In particular, the assumptions underlying the transformations are not always stated explicitly, and some of the mathematical steps seem to rely on implicit reasoning.

   o Eq. (1): I have downloaded the cited articles (Yuan et al. (2012) and Yang et al. (2019)), but no such equation as this was provided. I would recommend the authors to outline the derivation process, or at least to clarify the basic assumptions behind. Especially, how are the turbulence, wave orbital motions, and background current are separated, and what kind of assumptions are made about their timescales? How is the turbulent Reynolds stress modeled?

   o Eq. (12): This representation of mixing coefficient $\langle \bar{k}^2 / \pi^2 \bar{\varepsilon} \rangle_{SM}$ is not supported with the information provided so far. Please justify with a clear statement of underlying assumptions.

   o Eq. (19): Even if Eq. (13) is assumed to be valid, this expression appears to contain an error. In particular, it is unclear how the coefficient $7\sqrt{7}/16$ arises. Please clarify.

   o L348: Why can one transform like $\frac{c_0^2}{2g\mu_0^{1/2}} = \frac{g\mu_2}{2\mu_0^{1/2}\mu_4}$? It should be valid only when waves are monochromatic. It contradicts with later assumed $\rho = 0.4, 0.5, 0.6$ or so.

   o Eq. (26): There is no explanation or justification of this new expression. Please clarify what assumption brings Eq. (21) or (24) to this form. What does the integral term physically represent?

3. The comparison of growth and decay rates in L223-250:

   o To evaluate the decay rate, one needs to assume some value of $A$, but no explanation is provided. Please clarify.

   o L244-245: Without a relation between $u_*$ and $u_{w0}$, the comparison between two figures does not make sense. Also, which parameter range do "normal and extreme sea conditions" refer to?

   o Fig 2: The plotting range in $y$-axis, $0 \leq u_{w0} \leq 8$ m/s, seems to be comparable to, or exceed the phase speed of waves with wavenumbers shown here. Such a situation is unrealistic.

4. The comparison of measured TKE dissipation and wave KE dissipation in L286-310:

   o Through the analysis, do the authors try to argue that the TKE dissipation rate $\varepsilon_{dis}$ is locally balanced by the modeled wave-induced production $e_{tid}$? Such an argument requires quantitative discussion of other terms in the TKE equation because, in such a strongly forced short-fetched wind wave situation, neglected

terms such as the TKE production by breaking waves and wind-driven shear turbulence and the nonlocal transport of TKE through Langmuir turbulence are likely significant.

- o Fig 4 seems to contain an error. The slope of the model line should be proportional to $K = 2\pi/$wavelength, but the slopes of Exp.1 (red) and Exp.3 (blue) are nearly identical. According to Table 1, they should differ by a factor of about 2.

5. Section 3: The purpose and context are unclear.

- o The authors compare MASNUM results using different tuning parameters with the academic case presented in Janssen et al. (1994), but both are numerical model outputs. In this framework, it is not appropriate to refer to this as a "validation" (L416), since no observational or benchmark data are used for comparison, and there is no basis to judge which result is "better".

- o The quantitative behavior of the model can always be adjusted with tuning parameters. Comparing new model results (Experiments 2 and 3) without calibrating the parameters does not demonstrate that one model performs better than another.

6. Section 4: The discussion lacks a clear physical rationale and seems to drift away from the central point.

- o Here the authors argue that the parameter $\alpha_{wt}$ should depend on the orbital-velocity-based gradient Richardson number $Ri_g$. In Eq. (2), $\alpha_{wt}$ is originally introduced as the ratio of phase-averaged TKE production to that evaluated with phase-averaged eddy viscosity and strain rate. It would be helpful if the authors could clarify how the stratification (i.e., $Ri_g$) can affect the phase-dependent variation of turbulence quantities and thereby modify $\alpha_{wt}$.

- o Moreover, the stability criterion based on the gradient Richardson number is only a necessary condition for instability and should not be interpreted as a sufficient condition suggesting that "the perturbation must be amplified" (L504). Accordingly, the evaluations in Table 3 and Figure 10 do not ensure that the areas satisfying the criterion are turbulent. It also remains unclear how this evaluation is relevant to determining $\alpha_{wt}$.

7. Section 5: Likewise in Abstract, the conclusion seems to overstate the outcome of the present study. For example, in the paragraph starting at L543, the authors suggest that the wave dissipation with breaking only is insufficient based on the experiment made in Section 3. Without calibrating the tuning coefficients, even if one setup performs better than another, it does not necessarily mean the former represents physical processes more realistically. Furthermore, the new formulation was compared against

another simulation (Janssen et al., 1994), which cannot necessarily be considered as reference.

**Specific comments and minor issues**

8.  L4: I do not understand what "high-deterministic model" refers to.
9.  There are many undefined mathematical symbols throughout the manuscript: $\sigma_0, K_0, K, k_1, k_2, c_0, T_z$ etc. Since similar symbols are used to represent totally different quantities in this manuscript (e.g., $L$ vs $\bar{L}$), clear definitions are necessary.
10. L157 I suppose $k$ should represent TKE, not $k^2$.
11. L162 KE and PE seem to have inconsistent dimensions.
12. L195 $\langle AA^* \rangle = \delta(\vec{k} - \vec{k}')E(k_1, k_2)$ is incorrect. Delta function should arise as a result of integration of $\exp i(k - k')x$.
13. Eq. (9) It is unclear why the authors need this transformation (multiplying the numerator and denominator by $\iint_{\vec{k}} \omega^2 K^2 E(k_1, k_2) dk_1 dk_2$), as it is not used later.
14. L243 Define $\tau_w$ and $\tau$.
15. L243-244: Is there a reasonable explanation for the choice of these parameters $\tau_w/\tau$ and $\alpha_{wt}$?
16. L273 "Layer depth $x_3$ varies corresponding to different $A$" How and why?
17. L335 $\varepsilon$ is used for TKE dissipation in Section 2. I would recommend using another character.
18. L401-408 There is no explanation on the simulated domain (both in spatial and spectral).

---

## Author Comment (AC1)

**Response on RC1**

Meng Sun (correspondent author)

September 16, 2025

First and foremost, the authors want to express our sincere gratitude to the reviewer for taking the time to review our manuscript titled "Analytical approaches for wave energy dissipation induced by wave-generated turbulence and random wave-breaking". Your thoughtful feedback and constructive suggestions have been immensely valuable in improving the quality of our work.

The authors thank the reviewer for the overall very comprehensive summary of our work。 And we fully agree with the reviewer's suggestion that our manuscript should be edited by native English speakers.

In Section 2.1, we attempt to detect the scales of the proposed dissipation source function induced by wave-generated turbulence in Eq. (15) by comparing to wave growth formulations, and verify the modeled TKE dissipation rate, generated by shear instability of irregular wind waves or swells, with laboratory observations (Figs. 3 and 4). We entirely approve of the reviewer's point that the input mechanisms and in particular bulk transfer of energy action and momentum to waves is not well established either. Figures 1 and 2 in our manuscript show the growth/dissipation rate between input and dissipation terms under normal and extreme sea conditions, though both are comparable in spatial distribution and magnitude, and the growth is dominated by spectral signatures of the difference definitely. There is lack of comparison and assessment of Eq. (15) with previous formulations in the manuscript, which will be pursued in our future study. To further note, Figure 7 in Section 3 shows obliquely the role of Eq. (15) , comparable to that of previous parameterizations due to wave-breaking in the WAM and MASNUM wave models.

In Section 2.2, we present a new dissipation coefficient (26) based on the basic statistics of wave breaking by Yuan et al. (2009) and follow-up inspections and verifications (Wang et al., 2017, 2018; Shi et al., 2025). For easily displaying its relation to the previous formulation (24) (Yuan et al., 1986; Yuan et al.,1993; Donelan and Yuan, 1994), the new dissipation coefficient is rewritten as (27) by introducing the ratio of the kinetic energy loss to the potential one due to wave-breaking. Comparison to the previous formulation (24) indicates the new one varies with the ratio of the kinetic energy loss to the potential one due to wave-breaking, and it aligns with the previous one under certain conditions, while the constant coefficient in the previous one comes originally from the complicated 0-1 $^{st}$ order asymptotic expansions of the covariance of surface elevation. That is to say, it is more reasonable physically to introduce the kinetic energy loss and the potential one to the dissipation coefficient. While this theoretical argument indicates the improvement of the new dissipation coefficient to the previous one, we agree that it is not a direct comparison which needs our further study. And we apologize for our ambiguous expressions for lacking clarity and will rephrase and/or extend to improve the readability of the text. We also agree that there is considerable noise for the shorter wave scales in the observation data in Figs. 5

and 6, which Young and Babanin (2006) also stated. Polnikov (2012) also criticized the observation data of "inevitable statistical noise", we think the observation data still impart the fact that the longer wave scales are more affected by the dominant breaking than the shorter wave scales. Besides, as we know until now, it is the only in-lake site measurement to obtain the valuable breaking spectra and the nonbreaking spectra for the spectral difference. Comparisons of the attenuation coefficients by using Eq. (26) with the ratio of the two observed spectra may yield valuable insights, which also need further interpretations.

Key references

Donelan, M. A., and Yuan, Y.: Wave dissipation by surface processes, in: Dynamics and Modelling of Ocean Waves, edited by: Komen, G. J., Cavaleri, L., Donelan, M.,Hasselmann, K., Hasselmann, S., and Janssen, P. A. E. M., Cambridge University Press, Cambridge, UK, 143-155, ISBN 0-521-47047-1, 1994.

Yuan, Y., Han, L., Hua, F., Zhang, S., Qiao, F., Yang, Y., and Xia, C.: The statistical theory of breaking entrainment depth and surface whitecap coverage of real sea waves, J. Phys. Oceanogr., 39, 143-161, https://doi.org/10.1175/2008JPO3944.1, 2009.

Wang, H., Yang, Y., Sun, B., and Shi, Y.: Improvements to the statistical theoretical model for wave breaking based on the ratio of breaking wave kinetic and potential energy, Sci. China Earth Sci., 60(1), 180-187, https://doi.org/10.1007/s11430-016-0053-3, 2017.

Wang, H., Yang, Y., Dong, C., Su, T., Sun, B., and Zou, B.: Validation of an improved statistical theory for sea surface whitecap coverage using satellite remote sensing data, Sensors, 18, 3306, https://doi.org/10.3390/s18103306, 2018.

Shi, Y, Yang, Y., Qi, J., and Wang, H.: Adaptability assessment of the whitecap statistical physics model with cruise observations under high sea states, Front. Mar. Sci. 12:1486860, https://doi.org/10.3389/fmars.2025.1486860, 2025.

Young, I. R., and Babanin, A. V.: Spectral distribution of energy dissipation of wind-generated waves due to dominant wave breaking, J. Phys. Oceanogr., 36, 376-394, https://doi.org/10.1175/JPO2859.1, 2006.

Polnikov, V.G.: Spectral description of the dissipation mechanism for wind waves. Eddy viscosity model, Mar. Sci., 2(3), 13-26. https://doi.org/10.5923/j.ms.20120203.01., 2012

In order to evaluate the new dissipation formulations due to wave-generated turbulence and wave-breaking, which are focal points of the present study, the scaling behavior only for the duration-limited growth and decay is demonstrated and interpreted preliminary in section 3. Simple numerical experiments were performed and model results were compared to the original MASNUM wave model and WAM wave model (Janssen et al., 1994). We agree the reviewer' opinion that this is the weakest part of the manuscript and the listed 4 reasons in Rcs.

1. For duration-limited growth, no good observations lead us to compare to the original MASNUM wave model and WAM wave model (Janssen et al., 1994), to evaluate the

stable and reliable performance of the new dissipation formulations, and furthermore, to perceive directly the different effects between the two new dissipation formulations due to wave-generated turbulence and wave-breaking.

2. We agree the reviewer's opinion that it is possible to tune the models to get similar results by using free tuning parameters. Indeed in the new proposed dissipation formulations, there are still two undetermined parameters. Their valid ranges are inferred individually from the independent observation verifications. We discussed more concentrately about the two undetermined parameters in "Section 4 Dicussions". We think we pursued an attempt in numerical modeling not to tune parameters freely but to inspect their physics based on observations.

3. We thank the reviewer's advice that matching the new physical model with fetch-limited data associated with wave growth. In fact, we are conducting such studies for uniform wind, turning wind and rotatory wind in fetch-limited conditions which are also stated in Section 5. These model results and verifications with abundant fetch-limited observation data will be part of scaling behaviors in our future series papers (also due to the length limit of the journal). Here in the present study, we focus on the effects of the new dissipation formulations due to wave-generated turbulence and wave-breaking. For conditions of wind seas into swells, we agree the reviewer's point that different models exhibit different behavior about the decline of wave height. The different effects between the two new dissipation formulations are of concern to us and to manifest explicitly in this study.

4. We thank the reviewer's advice that it is better to show the spectral shapes, it will be pursued in our future project and displayed in other papers.

In Section 4, we concentrate on the undetermined parameters in the new dissipation formulations due to wave-generated turbulence and wave-breaking, including their physical meanings, respective origins, underlying challenges and possible future solutions. Although their valid ranges are inferred individually from independent observation verifications in sections 2 and 3, they are still the remain uncertainty issues which concern the main topics of the present study. The other purpose we discussed the undetermined parameters is to propose a potential way not to tune free parameters, which was commonly used previously for dissipation terms to balance the input ones in numerical modeling. We agree the reviewer's constructive suggestions to supplement the valuable comments and discussions provided above to Section 4, especially the deficiencies of scaling behavior of the new model. In the beginning of Section 4, we supplement these deficiencies as follows:

The analytical approaches and the corresponding comparisons to laboratory or in-lake site measurements improve further understandings of wave energy dissipation due to wave-breaking and wave-generated turbulence. This study still exhibits some deficiencies and needs to be addressed on comparative assessments and metrics to previous formulations, as well as evaluations of scaling behavior of the new model, etc. Model validation is tentative and requires future enhanced observations correspondingly.

We apologize for our ambiguous and confused expressions for lacking clarity and rigor in section 5. Following the reviewer's comments, we rewrote some paragraphs in this section.

1. We rewrote the first paragraph to "The ocean wave energy dissipation is the least understood of the major source terms, previous approaches to estimate the dissipation source function depended on an incomplete description of the physics of the processes including wave-breaking and wave-turbulence interaction. The latest observational efforts offer a possible approach to explore the underlying comprehensive mechanisms.' Some ambiguous words and subjective words are deleted according to the reviewer's suggestions.

2. We agree with the reviewer's opinion, and we rewrote the second paragraph to "In the present paper, we attempted to explore the dissipation effects of wave-generated turbulence reacting on ocean waves, and to estimate the energy loss due to wave-breaking via an improved postbreaking spectrum expression based on the breaking wave statistical method. Two new source functions for the above two dissipation processes are proposed and compared respectively to the laboratory or in-lake observations tentatively in section 2, and their different dissipation effects are experimentally analyzed in section 3."

3. As per the reviewer's suggestions, all subjective descriptors were removed to ensure academic rigor and appropriateness. The third paragraph is rewritten as "The main conclusion of the study is that we propose an analytical dissipation source function induced by wave-generated turbulence $S_{tid}$ formulated by Eq. (15), together with an improved postbreaking spectrum expression $E_b(k_1,k_2)$ by Eqs. (20) and (26). The former dissipation term represents the feedback of imparting of wave shear instability generations on turbulence, and the latter expression depicts the intermittent wave-breaking events. Sum of both contributions play critical role of wave energy dissipation."

4. We fully agree with the reviewer's opinion that there are uncertainties in wave growth formulations, as well as large uncertainties remained in different bulk energy input approaches stated in section 2.1. So in section 3, only the linear and quasi-linear wind input source functions are used for simplicity. Although the valid ranges of the two undetermined parameters in the new dissipation formulations are inferred individually from independent observation verifications, their certainty is still a significant challenge which we also discussed in section 4. So we are currently unable to conclusively address the reviewer's concerns about the overestimation of the input source term. We think we proposed a potential approach in this regard, but requires future enhanced observations correspondingly.

5. We thank the reviewer's advice and we are conducting such studies which will be part of our future series papers.

Some additional revisions are listed below, following the guidance of the reviewer:

Line 19: We rewrote the sentence to "Though input mechanisms, in particular bulk transfer of energy action and momentum to waves, and other source terms are not well established either, the least understood aspect of the physics of wave model is the dissipation terms (Donelan and Yuan, 1994; Young and Babanin, 2006; Babanin, 2011)".

Line 35-50: As per the reviewer's suggestions, the description and foundational journal references to WW3 were supplemented in these lines.

Based on the random phase spectral action density balance equation for wavenumber-direction spectra and evolved from earlier WW1 and WW2 (WAVEWATCH I & II) model packages (Tolman, 1991, 1992), the comprehensive source terms were incorporated and employed in WW3 ((WAVEWATCH III) wave model by Tolman and Chalikov (1996), Chalikov and Belevich (1993), Chalikov (1995) , Tolman (2002), etc.

Key references

Chalikov, D.: The parameterization of the wave boundary layer, J. Phys. Oceanogr., 25, 1,333-1,349 , 1995.

Chalikov, D. V., and Belevich, M. Y.: One-dimensional theory of the wave boundary layer, Bound.-Layer Meteor., 63, 65-96, 1993.

Tolman, H. L.: A third-generation model for wind waves on slowly varying, unsteady and inhomogeneous depths and currents, J. Phys. Oceanogr., 21, 782-797, 1991.

Tolman, H. L.: Eff ects of numerics on the physics in a third-generation wind-wave model, J. Phys. Oceanogr., 22, 1,095-1,111, 1992.

Tolman, H. L.: Validation of WAVEWATCH III version 1.15 for a global domain, Tech. Note 213, NOAA/NWS/NCEP/OMB, 33 pp., 2002.

Tolman, H. L., and Chalikov, D.: Source terms in a third-generation wind-wave model, J. Phys. Oceanogr., 26, 2497-2518, 1996.

Lines 71-72: We apologize for our inappropriate description of Tolman and Chalikov's work, we rewrote the sentence to "Tolman and Chalikov (1996) also suggested a turbulent dissipation analogy for tunable closure modeling."

Lines 79-83: The "third frequency range" here means the smooth transition between the low- and high-frequency dissipation mechanisms. We are grateful to the reviewer for spotting this, and we remove the sentence for it is not highly relevant to the main topics of the present study.

Thanks for your advice and we apologize for our ambiguous expressions lacking clarity of upper-case $K$ and lower-case $k$, upper case $K$ hat, $K_0$ , etc. in derivations leading up to Eq. (15) on line 220 . We supplemented their definitions in the revised manuscript respectively.

For the unified mean $\hat{\omega}, \hat{K}$ introduced for various integral mean variables for practical numerical applications, which satisfy

$$\iint_k E(k_1,k_2)\exp\{2Kx_3\}\mathrm{d}k_1\mathrm{d}k_2 \approx \exp\{2\hat{K}_1x_3\}\iint_k E(k_1,k_2)\mathrm{d}k_1\mathrm{d}k_2 \,, \iint_k \omega^2K^2E(k_1,k_2)\exp\{2Kx_3\}\mathrm{d}k_1\mathrm{d}k_2 \approx \hat{\omega}^2\hat{K}_2^2\exp\{2\hat{K}_2x_3\}\iint_k E(k_1,k_2)\mathrm{d}k_1\mathrm{d}k_2 \quad \text{and}$$

$$\iint_k \frac{1}{2K+2\hat{K}_1+\hat{K}_2}\omega^2K^2E(k_1,k_2)\mathrm{d}k_1\mathrm{d}k_2 \approx \frac{1}{5\hat{K}_3}\iint_k \omega^2K^2E(k_1,k_2)\mathrm{d}k_1\mathrm{d}k_2 \,, \text{ here we assume that } \hat{K}_1 \approx \hat{K}_2 \approx \hat{K}_3 \approx \hat{K}$$

approximately.

Lines 409-411: Two input source terms were used with the new dissipation source terms. One is the parameterization of linear growth in spectral density proposed by Komen et al. (1984), scaled in terms of the friction velocity rather than the wind speed at 5 m height which was adopted by Snyder et al. (1981). The other is the parameterization of quasi-linear one by Janssen (1991). We apologize for our ambiguous expressions to mislead the readers, We remove the reference of Snyder et al. (1981) for clarity.

---

## Author Comment (AC2)

**Response on RC2**

Meng Sun (correspondent author)
September 16, 2025

First and foremost, the authors want to express our sincere gratitude to the reviewer for taking the time to review our manuscript titled "Analytical approaches for wave energy dissipation induced by wave-generated turbulence and random wave-breaking". We are grateful for your thoughtful feedback and constructive suggestions in improving the quality of our work.

The authors thank the reviewer for the overall very comprehensive summary of our work。 And we fully agree with the reviewer's suggestion that our manuscript should be edited by a native English speaker or a professional editor.

We apologize for our ambiguous expressions for lacking clarity and will rephrase, extend and supplement assumptions and some critical derivation steps to improve the manuscript. In order to evaluate the new dissipation formulations due to wave-generated turbulence and wave-breaking, which are focal points of the present study, the numerical tests in section 3 consists of scaling behavior only for the duration-limited growth and decay. Simple numerical experiments were performed and model results were compared to the original MASNUM wave model and WAM wave model (Janssen et al., 1994). For duration-limited growth, no good observations lead us to compare to the original MASNUM wave model and WAM wave model (Janssen et al., 1994), to evaluate the stable and reliable performance of the new dissipation formulations, and furthermore, to perceive directly the different effects between the two new dissipation formulations due to wave-generated turbulence and wave-breaking. In addition, in our series of works, Wang et al. (2024) used the new model for global hindcast and forecast to evaluate the modeling performance by comparing to the Jason-3 satellite altimeter observation data and four non-nearshore NDBC buoys data. We are also conducting studies for uniform wind, turning wind and rotatory wind in fetch-limited conditions which are stated in Section 5. These model results and verifications with abundant fetch-limited observation data will be part of scaling behaviors in our future series papers (due to the length limit of the journal).

Key references

Wang, F., Yang, Y., Yin, X., Jiang, X., and Sun, M.: Improving wave modeling performance by incorporating wave-generated turbulence dissipation and improved post-breaking spectrum, Ocean Modell., 188 (2024) 102311, https://doi.org/10.1016/j.ocemod.2023.102311, 2024.

**Some major revisions are listed below, following the guidance of the reviewer.**

1.  Abstract

o   L6: Here "improved" means the new dissipation coefficient (26) that we present in section 2.2, For easily displaying its relation to the previous formulation (24) (Yuan et al., 1986;Yuan et al.,1993; Donelan and Yuan, 1994), the new dissipation coefficient is rewritten as

(27) by introducing the ratio of the kinetic energy loss to the potential one due to wave-breaking, which was first proposed by Yuan et al.(2009), inspected by Wang et al. (2017, 2018) and validated by Shi et al. (2025). Comparison to the previous one (24) indicates the new one varies with the ratio of the kinetic energy loss to the potential one due to wave-breaking, and it aligns with the previous one under certain conditions. The constant coefficient in the previous one comes originally from the complicated 0-1 $^{st}$ order asymptotic expansions of the covariance of surface elevation, which served as the foundation of the wave-breaking dissipation source function of the original MASNUM wave model. That is to say, it is more reasonable physically to introduce the kinetic energy loss and the potential one to the dissipation coefficient. We apologize for our ambiguous expressions for lacking clarity and will rephrase and/or extend to improve the text.

Key references

Donelan, M. A., and Yuan, Y.: Wave dissipation by surface processes, in: Dynamics and Modelling of Ocean Waves, edited by: Komen, G. J., Cavaleri, L., Donelan, M.,Hasselmann, K., Hasselmann, S., and Janssen, P. A. E. M., Cambridge University Press, Cambridge, UK, 143-155, ISBN 0-521-47047-1, 1994.

Yuan, Y., Han, L., Hua, F., Zhang, S., Qiao, F., Yang, Y., and Xia, C.: The statistical theory of breaking entrainment depth and surface whitecap coverage of real sea waves, J. Phys. Oceanogr., 39, 143-161, https://doi.org/10.1175/2008JPO3944.1, 2009.

Wang, H., Yang, Y., Sun, B., and Shi, Y.: Improvements to the statistical theoretical model for wave breaking based on the ratio of breaking wave kinetic and potential energy, Sci. China Earth Sci., 60(1), 180-187, https://doi.org/10.1007/s11430-016-0053-3, 2017.

Wang, H., Yang, Y., Dong, C., Su, T., Sun, B., and Zou, B.: Validation of an improved statistical theory for sea surface whitecap coverage using satellite remote sensing data, Sensors, 18, 3306, https://doi.org/10.3390/s18103306, 2018.

Shi, Y, Yang, Y., Qi, J., and Wang, H.: Adaptability assessment of the whitecap statistical physics model with cruise observations under high sea states, Front. Mar. Sci. 12:1486860, https://doi.org/10.3389/fmars.2025.1486860, 2025.

o L8: We agree that there is considerable noise in the observation data in Figs. 5 and 6, which Young and Babanin (2006) also stated, Some Comparisons may yield valuable insights and needs further interpretations. This sentence is changed to: "Their comparisons with laboratory observations or comprehensive measurements were provided and validated tentatively, and applications on simple duration-limited growth and decay experiments were implemented."

Key references

Young, I. R., and Babanin, A. V.: Spectral distribution of energy dissipation of wind-generated waves due to dominant wave breaking, J. Phys. Oceanogr., 36, 376-394, https://doi.org/10.1175/JPO2859.1, 2006.

2. The analytical derivation processes in section 2

○ Eq. (1): We apologize for a discrepancy in the cited date of the reference "Yang et al. (2019)", which should be revised to Yang et al. (2022). Equation (1) was provided in the latter reference. . As per the reviewer's suggestions, the description to outline the derivation process is supplemented briefly in Appendix A of the revised manuscript.

**Appendix A**

In the comprehensive framework of the ocean dynamic system comprised of wave-like motions, eddy-like motions and circulation, which are controlled by dynamic gravity balance, static gravity balance and geotropic balance respectively, ocean turbulence, highly random perturbations due to strong nonlinear advections in the foregoing three sub-systems, interacts with larger scale motions including the advection transport and shear instability generation of large-scale dynamic processes as well as the mixing effect in the form of its transport flux residual on the latter (Yuan et al., 2012). More comprehensive governing equations for wave motion were derived by Yuan et al. (2012) and formulated in tensor expression as follows:

$$\frac{\partial u_{\mathrm{SM}j}}{\partial x_j} = 0 , \tag{A1}$$

$$\begin{aligned} &\frac{\partial u_{\mathrm{SM}i}}{\partial t} + \hat{U}_j \frac{\partial u_{\mathrm{SM}i}}{\partial x_j} + u_{\mathrm{SM}j}\frac{\partial \hat{U}_i}{\partial x_j} + \frac{\partial}{\partial x_j}\left(u_{\mathrm{SM}j}u_{\mathrm{SM}i} - \left\langle u_{\mathrm{SM}j}u_{\mathrm{SM}i}\right\rangle_{\mathrm{SM}}\right) - 2\varepsilon_{ijk}u_{\mathrm{SM}j}\Omega_k \\ &= -\frac{1}{\rho_0}\frac{\partial p_{\mathrm{SM}}}{\partial x_i} - g\frac{\rho_{\mathrm{SM}}}{\rho_0}\delta_{3i} + \frac{\partial}{\partial x_j}\left(v_0\frac{\partial u_{\mathrm{SM}i}}{\partial x_j}\right) + \frac{\partial}{\partial x_j}\left[-\left(\left\langle u_{\mathrm{SS}j}u_{\mathrm{SS}i}\right\rangle_{\mathrm{SS}} - \left\langle\left\langle u_{\mathrm{SS}j}u_{\mathrm{SS}i}\right\rangle_{\mathrm{SS}}\right\rangle_{\mathrm{SM}}\right)\right] \end{aligned}, \tag{A2}$$

$$\begin{aligned} &\frac{\partial \rho_{\mathrm{SM}}}{\partial t} + \hat{U}_j \frac{\partial \rho_{\mathrm{SM}}}{\partial x_j} + u_{\mathrm{SM}j}\frac{\partial \hat{\rho}}{\partial x_j} + \frac{\partial}{\partial x_j}\left(u_{\mathrm{SM}j}\rho_{\mathrm{SM}} - \left\langle u_{\mathrm{SM}j}\rho_{\mathrm{SM}}\right\rangle_{\mathrm{SM}}\right) \\ &= \frac{\partial}{\partial x_j}\left(K_0\frac{\partial \rho_{\mathrm{SM}}}{\partial x_j}\right) + \frac{\partial}{\partial x_j}\left[-\left(\left\langle u_{\mathrm{SS}j}\rho_{\mathrm{SS}}\right\rangle_{\mathrm{SS}} - \left\langle\left\langle u_{\mathrm{SS}j}\rho_{\mathrm{SS}}\right\rangle_{\mathrm{SS}}\right\rangle_{\mathrm{SM}}\right)\right] + Q_{\rho\mathrm{SM}} \end{aligned}, \tag{A3}$$

where $u_{\mathrm{SM}i}, i = 1, 2, 3; \rho_{\mathrm{SM}}$ denote the ocean wave components , $\hat{U}_i, i = 1,2,3; \hat{T}, \hat{s}, \hat{p}, \hat{\rho}$

denote the background current components , $\rho_0$ is the basin mean water density;

$v_0, K_0, D_0$ denote the molecular viscosity, thermal and diffusion coefficients; $Q_{\rho\mathrm{SM}}$

denotes the thermal source due to temperature and salinity perturbations; $\left\langle \cdot \right\rangle_{\mathrm{SS}}$, $\left\langle \cdot \right\rangle_{\mathrm{SM}}$

denote the Reynolds averages on the turbulence and wave motions respectively. Hereafter, other symbols have their usual meaning.

Multiplying Eq. (A2) by $\rho_0 u_{\mathrm{SM}i}$ and Eq. (A3) by $\rho_{\mathrm{SM}}$, after some manipulation,

yields the unit volume mechanical kinetic energy and potential energy respectively. Sum of both energy terms and Reynolds averaged on the wave motion reduce to

$$\frac{\partial}{\partial t}\left\langle\frac{\rho_0 u_{\mathrm{SM}i}^2}{2}+\frac{g^2\rho_{\mathrm{SM}}^2}{2\rho_0\hat{N}_3^2}\right\rangle_{\mathrm{SM}}+\left\langle\left(\hat{U}_j+u_{\mathrm{SM}j}\right)\frac{\partial}{\partial x_j}\left(\frac{\rho_0 u_{\mathrm{SM}i}^2}{2}+\frac{g^2\rho_{\mathrm{SM}}^2}{2\rho_0\hat{N}_3^2}\right)\right\rangle_{\mathrm{SM}}+\frac{\partial}{\partial x_i}\left\langle p_{\mathrm{SM}}u_{\mathrm{SM}i}\right\rangle_{\mathrm{SM}}$$

$$=-\rho_0\left\langle u_{\mathrm{SM}i}u_{\mathrm{SM}j}\right\rangle_{\mathrm{SM}}\frac{\partial\hat{U}_i}{\partial x_j}+g\left\langle\rho_{\mathrm{SM}}u_{\mathrm{SM}\beta}\right\rangle_{\mathrm{SM}}\frac{\hat{N}_\beta^2}{\hat{N}_3^2}+\frac{g^2}{\rho_0\hat{N}_3^2}\left\langle\rho_{\mathrm{SM}}Q_{\rho\mathrm{SM}}\right\rangle_{\mathrm{SM}}$$

$$+\left[\begin{array}{l}\left\langle\rho_0 u_{\mathrm{SM}i}\dfrac{\partial}{\partial x_j}\left(\nu_0\dfrac{\partial u_{\mathrm{SM}i}}{\partial x_j}\right)\right\rangle_{\mathrm{SM}}+\left\langle\dfrac{g^2\rho_{\mathrm{SM}}}{\rho_0\hat{N}_3^2}\dfrac{\partial}{\partial x_j}\left(K_0\dfrac{\partial\rho_{\mathrm{SM}}}{\partial x_j}\right)\right\rangle_{\mathrm{SM}}\\[4mm]+\left\langle\rho_0 u_{\mathrm{SM}i}\dfrac{\partial}{\partial x_j}\left(-\left\langle u_{\mathrm{SS}j}u_{\mathrm{SS}i}\right\rangle_{\mathrm{SS}}\right)\right\rangle_{\mathrm{SM}}+\left\langle\dfrac{g^2\rho_{\mathrm{SM}}}{\rho_0\hat{N}_3^2}\dfrac{\partial}{\partial x_j}\left(-\left\langle u_{\mathrm{SS}j}\rho_{\mathrm{SS}}\right\rangle_{\mathrm{SS}}\right)\right\rangle_{\mathrm{SM}}\end{array}\right]\quad, \qquad (A4)$$

Based on closure assumptions of the second-order turbulence moments equations with high certainty (Baumert et al., 2005; Yuan et al., 2013), Eq. (A4) can be written as

$$\frac{\partial}{\partial t}\left\langle\frac{\rho_0 u_{\mathrm{SM}i}^2}{2}+\frac{g^2\rho_{\mathrm{SM}}^2}{2\rho_0\hat{N}_3^2}\right\rangle_{\mathrm{SM}}+\left\langle\left(\hat{U}_j+u_{\mathrm{SM}j}\right)\frac{\partial}{\partial x_j}\left(\frac{\rho_0 u_{\mathrm{SM}i}^2}{2}+\frac{g^2\rho_{\mathrm{SM}}^2}{2\rho_0\hat{N}_3^2}\right)\right\rangle_{\mathrm{SM}}+\frac{\partial}{\partial x_i}\left\langle p_{\mathrm{SM}}u_{\mathrm{SM}i}\right\rangle_{\mathrm{SM}}$$

$$=\left(-\rho_0\left\langle u_{\mathrm{SM}i}u_{\mathrm{SM}j}\right\rangle_{\mathrm{SM}}\frac{\partial\hat{U}_i}{\partial x_j}+g\left\langle\rho_{\mathrm{SM}}u_{\mathrm{SM}\alpha}\right\rangle_{\mathrm{SM}}\frac{\hat{N}_\alpha^2}{\hat{N}_3^2}\right)+\frac{g^2}{\rho_0\hat{N}_3^2}\left\langle\rho_{\mathrm{SM}}Q_{\rho\mathrm{SM}}\right\rangle_{\mathrm{SM}} \qquad (A5)$$

$$+\left\{\begin{array}{l}\dfrac{\partial}{\partial x_j}\left\langle\left(\nu_0+\dfrac{k^2}{\pi^2\varepsilon}\right)\dfrac{\partial}{\partial x_j}\left(\dfrac{\rho_0 u_{\mathrm{SM}i}^2}{2}\right)\right\rangle_{\mathrm{SM}}+\dfrac{\partial}{\partial x_j}\left\langle\left(K_0+\dfrac{1}{\sigma_0}\dfrac{k^2}{\pi^2\varepsilon}\right)\dfrac{\partial}{\partial x_j}\left(\dfrac{g^2\rho_{\mathrm{SM}}^2}{2\rho_0\hat{N}_3^2}\right)\right\rangle_{\mathrm{SM}}\\[4mm]-\left\langle\rho_0\left(\nu_0+\dfrac{k^2}{\pi^2\varepsilon}\right)\left(\dfrac{\partial u_{\mathrm{SM}i}}{\partial x_j}\right)^2\right\rangle_{\mathrm{SM}}-\left\langle\dfrac{g^2}{\rho_0\hat{N}_3^2}\left(K_0+\dfrac{1}{\sigma_0}\dfrac{k^2}{\pi^2\varepsilon}\right)\left(\dfrac{\partial\rho_{\mathrm{SM}}}{\partial x_j}\right)^2\right\rangle_{\mathrm{SM}}\end{array}\right\}$$

where $\hat{N}_i^2=-g\dfrac{\partial}{\partial x_i}\left(\dfrac{\hat{\rho}}{\rho_0}\right)$, $i=1,2,3$ denote the Brunt–Väisälä frequency components; $k$, $\varepsilon$

the kinetic energy and its dissipation rate of ocean turbulence, which is generated by shear instability of background current, Stokes drift and wave orbital motions in the upper layers. Here in this study, only the wave-generated turbulence is considered. Its analytical mixing coefficients were proposed through equilibrium solutions of the second-order turbulence closure model between the wave motion shear instability generations and the TKE dissipations (Yuan et al., 2013).

Key references

Baumert, H. Z., Simpson, J. H., and Sündermann, J. (Eds.): Marine turbulence: theories, observations, and models, Cambridge University Press, Cambridge, UK, 630pp., ISBN 978-0-521-15372-0, 2005.

Yang, Y., Sun, M., Sun ,L., Xia, C., Teng, Y., and Cui, X.: A Characteristics Set Computation Model for Internal Wavenumber Spectra and Its Validation with MODIS Retrieved Parameters in the Sulu Sea and Celebes Sea, Remote Sens., 14, 1967, https://doi.org/10.3390/rs14091967, 2022.

Yuan, Y., Qiao, F., Yin, X., and Han, L.: Establishment of the ocean dynamic system with four sub-systems and the derivation of their governing equation sets, J. Hydrodyn., 24, 153-168, https://doi.org/10.1016/S1001-6058(11)60231-X, 2012.

Yuan, Y., Qiao, F., Yin, X., and Han, L.: Analytical estimation of mixing coefficient induced by surface wave-generated turbulence based on the equilibrium solution of the

second-order turbulence closure model, Sci. China Earth Sci., 56, 71-80,
https://doi.org/10.1007/s11430-012-4517-x , 2013.

○ Eq. (12): We add information, as well as underlying assumptions in the preceding paragraph of Eq. (12), and the paragraph is rewritten as

Yuan et al. (2013) proposed a parameterization of the mixing coefficient $\left\langle \dfrac{\overline{k}^2}{\pi^2 \overline{\varepsilon}} \right\rangle_{\mathrm{SM}}$ through equilibrium solutions of the second-order turbulence model with high certainty closure assumptions, in which the power function relationship between turbulent dissipation rate and shear instability generation of wave motion was fitted by observation data in deep ocean.

Here we choose a generic representation of the mixing length of $\overline{l}_D = \dfrac{\overline{k}^{3/2}}{\pi^{3/2} \overline{\varepsilon}}$ (Baumert et al., 2005), which is appropriate for deep and shallow water conditions, and the mixing coefficient is formulated conveniently as
Key references

Baumert, H. Z., Simpson, J. H., and Sündermann, J. (Eds.): Marine turbulence: theories, observations, and models, Cambridge University Press, Cambridge, UK, 630pp., ISBN 978-0-521-15372-0, 2005.

Yuan, Y., Qiao, F., Yin, X., and Han, L.: Analytical estimation of mixing coefficient induced by surface wave-generated turbulence based on the equilibrium solution of the second-order turbulence closure model, Sci. China Earth Sci., 56, 71-80, https://doi.org/10.1007/s11430-012-4517-x , 2013.

○ Eq. (19): There is no problem in Eq. (19), we carefully checked again. There is a clarification should be provided prior to Eq. (19) about an approximate coefficient 7/8. So the sentence and the equation are revised to
In consideration of an approximate coefficient 7/8 introduced from the minimization relation for the equilibrium solutions (Yuan et al., 2013), the TKE dissipation rate $\varepsilon_{\mathrm{dis}}$ can be derived as

$$\varepsilon_{\mathrm{dis}} \approx \alpha_{\mathrm{wt}} \frac{7}{8} \left\langle \frac{\overline{k}^2}{\pi^2 \overline{\varepsilon}} \right\rangle_{\mathrm{SM}} \left\langle \left( \frac{\partial u_{\mathrm{SM}i}}{\partial x_j} \right)^2 \right\rangle_{\mathrm{SM}} = \frac{7\sqrt{7}}{16} \alpha_{\mathrm{wt}} A^5 \omega^3 K^3 \exp\{5K x_3\} \tag{19}$$

For monochromatic non-breaking waves, the derivation processes are similar to those for wave spectrum in the manuscript. For deep water depth,

$$\left\langle \frac{\partial u_{\mathrm{SM}i}}{\partial x_j} \frac{\partial u_{\mathrm{SM}i}}{\partial x_j} \right\rangle_{\mathrm{SM}} = 2 A^2 \omega^2 K^2 \exp(2K x_3)$$

and

$$\left\langle \frac{\overline{k}^2}{\pi^2 \overline{\varepsilon}} \right\rangle_{SM} = \frac{\sqrt{7}}{4} A^3 \omega K \exp\{3Kx_3\}$$

(In fact, let $Hs = 2\sqrt{2}A$, here $A$ represents the root-mean-square wave amplitude, Eq. (14) for wave spectrum in the manuscript can be reduced to the above equation.) The two equations yield the above Eq. (19).

Key references

Yuan, Y., Qiao, F., Yin, X., and Han, L.: Analytical estimation of mixing coefficient induced by surface wave-generated turbulence based on the equilibrium solution of the second-order turbulence closure model, Sci. China Earth Sci., 56, 71-80, https://doi.org/10.1007/s11430-012-4517-x , 2013.

o   L348: We thanks for the reviewer's insightful comments. The transforms here are also valid approximately under a narrow spectrum assumption for theoretical arguments applied widely in the breaking wave statistics (Yuan et al., 1986, 2009; Donelan and Yuan, 1994). For practical numerical modeling, the wave spectrum is not actually narrow; thus, it is not precise enough to use on the results based on a narrow spectrum. We supplement the underlying assumption in the sentence.

Key references

Donelan, M. A., and Yuan, Y.: Wave dissipation by surface processes, in: Dynamics and Modelling of Ocean Waves, edited by: Komen, G. J., Cavaleri, L., Donelan, M.,Hasselmann, K., Hasselmann, S., and Janssen, P. A. E. M., Cambridge University Press, Cambridge, UK, 143-155, ISBN 0-521-47047-1, 1994.

Yuan, Y., Han, L., Hua, F., Zhang, S., Qiao, F., Yang, Y., and Xia, C.: The statistical theory of breaking entrainment depth and surface whitecap coverage of real sea waves, J. Phys. Oceanogr., 39, 143-161, https://doi.org/10.1175/2008JPO3944.1, 2009.

o   Eq.(26): We apologize for lacking clarity of the assumption and derivation process from Eq. (21) or (24) to Eq.(26). Yuan et al. (2009) derived some basic statistics of wave breaking for a narrow spectrum, especially the breaking kinetic and potential energy loss which add up to deduce the breaking mechanical energy loss formulated by introducing the ratio of the former to the latter. Wang et al. (2017, 2018) concluded that the ratio is mainly within the range 3-30, which indicates that there is a disproportion feature between the wave kinetic energy loss and potential one due to wave-breaking. Shi et al. (2025) validated the statistical wave-breaking model across multiple sites from the High Wind Speed Gas Exchange Study (HiWinGS), and concluded that the model is highly effective in capturing the dynamics of whitecap coverage across a range of high sea states. Based on the latest findings, an improved attenuation coefficient by introducing the breaking kinetic energy loss is proposed as follows:

$$\alpha_b' = 1 - \rho \frac{\omega^2 \mu_0^{1/2}}{g} \frac{1}{L} \left[ L \int_{-\infty}^{-L} \exp\left\{ -\frac{1}{2} Z^2 \right\} dZ + \exp\left\{ -\frac{\rho^2}{8} \frac{g^2}{\mu_0 \omega_z^4} \right\} \right] \qquad (26)$$

where the first and second terms in the bracket on the right-hand side are related to the dimensionless breaking kinetic and potential energy loss respectively,

$$L = \frac{g}{2\mu_4^{1/2}} = \frac{\rho}{\pi\lambda}\left(\frac{H_\mathrm{S}}{\overline{L}}\right)^{-1} = \frac{\rho}{2}\frac{g}{\mu_0^{1/2}\omega_z^2} \text{ , mean wavelength } \overline{L} = \lambda\frac{g}{2\pi}T_z^2 , \lambda = \frac{2}{3} \text{ or 0.86 (Kinsman,}$$

2012; Yuan et al., 2009; Xu and Yu, 2001).

Key references

Yuan, Y., Han, L., Hua, F., Zhang, S., Qiao, F., Yang, Y., and Xia, C.: The statistical theory of breaking entrainment depth and surface whitecap coverage of real sea waves, J. Phys. Oceanogr., 39, 143-161, https://doi.org/10.1175/2008JPO3944.1, 2009.

Wang, H., Yang, Y., Sun, B., and Shi, Y.: Improvements to the statistical theoretical model for wave breaking based on the ratio of breaking wave kinetic and potential energy, Sci. China Earth Sci., 60(1), 180-187, https://doi.org/10.1007/s11430-016-0053-3, 2017.

Wang, H., Yang, Y., Dong, C., Su, T., Sun, B., and Zou, B.: Validation of an improved statistical theory for sea surface whitecap coverage using satellite remote sensing data, Sensors, 18, 3306, https://doi.org/10.3390/s18103306, 2018.

Shi, Y, Yang, Y., Qi, J., and Wang, H.: Adaptability assessment of the whitecap statistical physics model with cruise observations under high sea states, Front. Mar. Sci. 12:1486860, https://doi.org/10.3389/fmars.2025.1486860, 2025.

3.  The comparison of growth and decay rates in L223-250:

o   The wave orbital velocity at sea surface $u_{\mathrm{w0}}$ varies with the wave amplitude $A$,

$u_{\mathrm{w0}} = A\hat{\omega}$ .The vertical axis in Fig. 2 represents the changes of $u_{\mathrm{w0}}$, which indicates that the

amplitude $A$ varies correspondingly.

o   L244-245: We agree with the reviewer's point that there is no rigorous relation between $u_*$ and $u_{\mathrm{w0}}$. For wind waves, the two variables exhibit a positive qualitative relationship. Furthermore, our main topic is to show the difference of the growth and dissipation rate between Fig.1 and Fig.2, for $u_*$ and $u_{\mathrm{w0}}$ varies in a range across normal and extreme sea conditions (the horizontal axis variable $K$ needs to be considered together).

o   Fig 2: We thanks for the reviewer's insightful comments. We redraw Fig. 2 again in consideration of the constraint. Figure 1 is also redrawn together.

[Figure]

**Figure 1**. Distribution of growth rate $\gamma$ as a function of wavenumber and friction velocity (Unit: $s^{-1}$)

[Figure]

**Figure 2**. Distribution of dissipation rate $\gamma_{tid}$ as a function of wavenumber and wave orbital velocity at sea surface (Unit: $s^{-1}$)

4. The comparison of measured TKE dissipation and wave KE dissipation in L286-310:

○ We agree with the reviewer's point that the TKE production by breaking waves, wind-driven shear turbulence and the nonlocal transport of TKE through Langmuir turbulence are likely significant for the TKE balance. In this study only the wave-generated turbulence is concerned for simplicity, in addition, the nonbreaking waves observed were unforced (no wind), mechanically generated, deep-water, two-dimensional wave trains in Fig. 3 and sources of the shear production were carefully eliminated, so the turbulence observed must have been directly generated by the waves themselves(Babanin and Haus, 2009). The TKE dissipation rate $\varepsilon_{dis}$ can be virtually identical to $e_{tid}$ numerically, but it should be noted that the two physical quantities have different physical meanings.

o   We apologize for the confusions between observations for experiment 1 and observations for experiment 2, between Line 1 with Line 2 of model results in Fig.4. Thanks for the reviewer's insightful comments, and we redraw the figure as follows:

[Figure]

**Figure 4**. Dependence of TKE dissipation rate $\varepsilon_{dis}$ (denoted as $\varepsilon$ in the figure) versus layer depth. Observation data (circles, pluses and triangles) are digitalized from Wei et al. (2018). Dependence (19) is shown with solid lines. Data are plotted in linear/logarithmic scale on the horizontal/vertical axis.

5.   Section 3: The purpose and context

o   We agree with reviewer's opinions that this section consists of scaling behavior only for the duration-limited growth and decay in order to evaluate the new dissipation formulations due to wave-generated turbulence and wave-breaking and it is not appropriate to refer to this as a "validation" (L416). So the sentence is revised to
To evaluate the scaling behavior of the new dissipation formulations due to wave-generated turbulence and wave-breaking proposed in Section 2, as well as their different effects, several numerical experiments are carried out (Table 2).

o   We agree the reviewer's opinion that it is possible to tune the models to get similar results by using free tuning parameters. Indeed in the new proposed dissipation formulations, there are still two undetermined parameters. Their valid ranges are inferred individually from the independent observation verifications. We discussed more concentrately about the two undetermined parameters in "Section 4 Dicussions". We think we pursued an attempt in numerical modeling not to tune parameters freely but to inspect their physics based on observations.

6.  Section 4:

o  In Section 4, we concentrate on the undetermined parameters in the new dissipation formulations due to wave-generated turbulence and wave-breaking, including their physical meanings, respective origins, underlying challenges and possible future solutions. Although their valid ranges are inferred individually from independent observation verifications in Section 2 and 3, they are still the remain uncertainty issues which concern the main topics of the present study. The other purpose we discussed the undetermined parameters is to propose a potential way not to tune free parameters, which was commonly used previously for dissipation terms to balance the input ones in numerical modeling. We agree the reviewer's constructive suggestions to clarify the effects of the gradient Richardson number $R_g$ on the

   turbulence and the modification to the $\alpha_{wt}$. In this study, the undetermined $\alpha_{wt}$ implies the

   quasi-equilibrium level of wave-generated turbulence, originated from the equilibrium solutions of the second-order turbulence model with high certainty closure assumptions. It is qualitatively related to the instability area proportion as shown in Fig. 10 or the instability volume proportion beneath the wave surface, albeit it does not ensure the areas satisfying the criterion of the critical value $R_g^c$ are turbulent. This is a challenging problem and our current arguments are still tentative.

o  We agree with the reviewer's opinions that he stability criterion based on the gradient Richardson number is only a necessary condition for instability and the evaluations do not ensure that the areas satisfying the criterion are turbulent. We remove the not-so-rigorous claim "the perturbation must be amplified" (L504) and supplement the qualitative

   relationship between the undetermined $\alpha_{wt}$ and the gradient Richardson number $R_g$ as:

   The coefficient $\alpha_{wt}$ is qualitatively related to the two-dimensional instability area

   proportion as shown in Fig. 10 or the instability volume proportion beneath the wave surface in real scenarios, which needs further studies.

7.  Section 5

In this study, our main topics are the analytical dissipation source functions induced by wave-generated turbulence and wave-breaking, and their validations by comparing to the independent laboratory or in-lake site measurements. For the latter, the valid range of the undermined parameter $\rho$ was inferred from validations of the sea surface whitecap courage obtained from the statistical wave-breaking model with the satellite-derived data (Wang et al. 2017, 2018), and its different effects are shown in Fig. 9. We agree with the reviewer's point this is still tentative and requires further elaboration, especially its challenge that we discussed in section 4. We modify the sentence at L543 rigorously to "Exploratory comparative analysis of the paper reveals that the wave energy loss only induced by wave-breaking appears to inadequate, which indicates that it may be somewhat overestimated in the previous studies." For duration-limited growth , no good observations lead us to compare to the original MASNUM wave model and

WAM wave model (Janssen et al., 1994), to evaluate the performance of the new dissipation formulations, and to perceive the different effects between the two new dissipation formulations due to wave-generated turbulence and wave-breaking. This is the weakest part of the manuscript, and we are conducting these works in fetch-limited conditions to compare with abundant fetch-limited observation data, which will be part of scaling behaviors in our future series papers.

**Specific comments and minor issues**

8.  We revised to "a second-order turbulence closure model between the wave shear instability generations and the turbulent kinetic energy (TKE) dissipations with high certainty closure assumptions" for terminological uniformity in the manuscript.

9.  We supplement the definitions of these mathematical symbols in the revised manuscript.

10. We apologize for the writing error for definition of the kinetic energy at L157, the power of 2 should be deleted.

11. We carefully check again the dimensions of KE and PE at L162, both have consistent dimensions, in consideration of the definition of the Brunt–Väisälä frequency

$$\hat{N}_3^2 = -g\frac{\partial}{\partial x_3}\left(\frac{\hat{\rho}}{\rho_0}\right) \quad \text{at L 157.}$$

12. This expression follows that in Chapter "7.3 THE WAVE SPECTRUM" of the book " Wind Waves" by Kinsman (2012). This expression is not entirely rigorous, but convenient for the following derivation processes, we adopt this simplified notation and there is no problem for the derived formulations.

    Kinsman, B. (Eds.): Wind Waves: Their Generation and Propagation on the Ocean Surface, Dover Publications, Inc., New York, USA, 676 pp., ISBN 978-0-486-64652-7, 2012.

13. this transformation is used to introduce the following unified mean $\hat{\omega}, \hat{K}$ for various integral mean variables, in order to obtain the convenient approximation (15) at L220.The unified mean $\hat{\omega}, \hat{K}$ are introduced for various integral mean variables for practical numerical applications, which satisfy

$$\iint_{\vec{k}} E(k_1,k_2)\exp\{2Kx_3\}\,dk_1dk_2 \approx \exp\{2\hat{K}_1x_3\}\iint_{\vec{k}} E(k_1,k_2)dk_1dk_2 \ , \ \iint_{\vec{k}} \omega^2 K^2 E(k_1,k_2)\exp\{2Kx_3\}\,dk_1dk_2 \approx \hat{\omega}^2\hat{K}_2^2\exp\{2\hat{K}_2x_3\}\iint_{\vec{k}} E(k_1,k_2)dk_1dk_2$$

and $\iint_{\vec{k}} \dfrac{1}{2K+2\hat{K}_1+\hat{K}_2}\omega^2 K^2 E(k_1,k_2)dk_1dk_2 \approx \dfrac{1}{5\hat{K}_3}\iint_{\vec{k}} \omega^2 K^2 E(k_1,k_2)dk_1dk_2$ . Here we assume that $\hat{K}_1 \approx \hat{K}_2 \approx \hat{K}_3 \approx \hat{K}$ approximately.

14. The definitions of $\tau_w, \tau$ at L243 are supplemented in the revised manuscript.

15. The wave-induced stress $\tau_w$, is less than the total stress $\tau = u_*^2$ with $u_*$ the friction velocity (Janssen 1991), so we set $\dfrac{\tau_w}{\tau} = 0.5$ for simplicity, which means the wave-induced stress accounts for a large proportion. $\alpha_{wt}$ implies the quasi-equilibrium level of wave-generated turbulence, here we set $\alpha_{wt} = 1$ which means a state of balance between the wave motion shear instability generations and the TKE dissipations is achieved (Yuan et al., 2013).

16. The significant TKE dissipation rates $\varepsilon_{dis}$ were retrieved from the measurements at the 30 mm layer from the still surface for all recorded nonbreaking waves, which is stated at L269. So the layer depth $x_3$ varies for different waves, and we explicitly account for this variable during our model estimation of the TKE dissipation rates $\varepsilon_{dis}$.

17. We thanks for the reviewer's advice and we change the parameter $\varepsilon$ at L335 and L337 to $\varepsilon_{sp}$.

18. The model covers a 23°×27° geographic region with 0.25°×0.25° horizontal resolution (The modeling spectral space was set as 24 directions with intervals of 15° and 25 wavenumbers spaced exponentially from the minimum wavenumber of 0.0071 m$^{-1}$ up to 0.6894 m$^{-1}$ with intervals of $K_i/K_{i-1}$=1.21, i=2,3,…,25). Model results in the middle of the geographic region are selected for numerical comparisons. In fact, the model exhibits spatially uniform performance for the simple duration-limited growth and decay experiments, except for a confined area near boundaries. So it appears to be not necessary to describe the model settings or the simulated domain in the manuscript. We will clarify them in our future studies in fetch-limited conditions, in which this issue is of critical importance.